# Improved Best-of-Both-Worlds Regret for Bandits with Delayed Feedback

**Ofir Schlisselberg**
Tel Aviv University
ofirs4@mail.tau.ac.il

**Tal Lancewicki**
Tel Aviv University
lancewicki@mail.tau.ac.il

**Peter Auer**
Technical University of Leoben
auer@unileoben.ac.at

**Yishay Mansour**
Tel Aviv University and Google Research
mansour.yishay@gmail.com

## Abstract

We study the multi-armed bandit problem with adversarially chosen delays in the Best-of-Both-Worlds (BoBW) framework, which aims to achieve near-optimal performance in both stochastic and adversarial environments. While prior work has made progress toward this goal, existing algorithms suffer from significant gaps to the known lower bounds, especially in the stochastic settings. Our main contribution is a new algorithm that, up to logarithmic factors, matches the known lower bounds in each setting individually.

In the adversarial case, our algorithm achieves regret of $\widetilde{O}(\sqrt{KT} + \sqrt{D})$, which is optimal up to logarithmic terms, where $T$ is the number of rounds, $K$ is the number of arms, and $D$ is the cumulative delay. In the stochastic case, we provide a regret bound which scale as $\sum_{i:\Delta_i>0}(\log(T)/\Delta_i) + \frac{1}{K}\sum \Delta_i \sigma_{max}$, where $\Delta_i$ is the sub-optimality gap of arm $i$ and $\sigma_{\max}$ is the maximum number of missing observations.

To the best of our knowledge, this is the first *BoBW* algorithm to simultaneously match the lower bounds in both stochastic and adversarial regimes in delayed environment. Moreover, even beyond the BoBW setting, our stochastic regret bound is the first to match the known lower bound under adversarial delays, improving the second term over the best known result by a factor of $K$.

## 1 Introduction

Delayed feedback presents a significant challenge that sequential decision-making algorithms encounter in many real-world applications. Notably, delays are often an inherent part of environments involving sequential decision-making, such as in healthcare, finance, and recommendation systems. As a central challenge in Online Learning, delays have been extensively explored in various contexts within Multi-armed Bandits (MAB), both in stochastic settings, where losses are generated i.i.d. from a fixed underlying distribution [18, 23, 38, 34, 35, 5, 12, 19, 36, 15, 29, 25] and adversarial settings, where the losses are chosen arbitrarily by an adversary [24, 4, 30, 2, 40, 16, 13, 32, 33].

Roughly speaking, under stochastic losses, delays contribute an additive regret term that does not scale with the time horizon (but with the number of missing observations), whereas under the adversarial losses, delays introduce an additive term that does scale with the horizon. More specifically, for an arbitrary sequence of delays, the best-known regret under stochastic losses is $\sum_{\Delta_i>0}\frac{\log(T)}{\Delta_i} + K\sigma_{max}$ (Joulani et al. [18]) where $T$ is the number of rounds, $K$ is the number of

39th Conference on Neural Information Processing Systems (NeurIPS 2025).

Table 1: Comparison of regret bounds (up to constants and $\log(K)$ factors) to the previous state-of-the-art regret both under stochastic and adversarial losses under adversarial delays.

| Algorithm | Regime | Regret |
|---|---|---|
| Joulani et al. [18] | stochastic | $\sum_{\Delta_i > 0}(\frac{\log(T)}{\Delta_i} + \sigma_{max}\Delta_i)$ |
| Thune et al. [30]
Zimmert and Seldin [40][2]
Gyorgy and Joulani [13][2] | adversarial | $\sqrt{TK} + \sqrt{D}$ |
| Masoudian et al. [22][2] | stochastic

adversarial | $\sum_{i:\Delta_i>0}(\frac{\log(T)}{\Delta_i} + \frac{\sigma_{max}}{\Delta_i}) + \Phi^*$
$\sqrt{TK} + \sqrt{D} + \Phi^* + K\sigma_{max}$
$\Phi^* = \min\left\{d_{max}K^{2/3}, \sqrt{DK^{2/3}}\right\}$ |
| Our paper[2] | stochastic
adversarial | $\sum_{i:\Delta_i>0}(\frac{\log(T)}{\Delta_i} + \sigma_{max}\frac{\Delta_i}{K})$
$\sqrt{TK\log(T)} + \sqrt{D}$ |
| Lower Bound
Lancewicki et al. [19] (constant delay)
Masoudian et al. [21][2] | stochastic
adversarial | $\sum_{i:\Delta_i>0}(\frac{\log(T)}{\Delta_i} + \sigma_{max}\frac{\Delta_i}{K})$
$\sqrt{TK} + \sqrt{D}$ |

arms, $\Delta_i$ is the sub-optimality gap of arm $i$ and $\sigma_{max}$ is the maximal number of missing observations. Under adversarial losses, the optimal bound is of the order $\sqrt{TK} + \sqrt{D}$ (Thune et al. [30], and later Zimmert and Seldin [40] and Gyorgy and Joulani [13] ), where $D$ is the sum of the delays.

**Remark 1** *The $\sqrt{D}$ term in the optimal regret corresponds to the worst-case delay. As shown, for example, in Zimmert and Seldin [40], Masoudian et al. [21, 22], Gyorgy and Joulani [14], this can be replaced with $\min_{S \subseteq [T]}\left\{|S| + \sqrt{D_{\bar{S}}}\right\}$, where $D_{\bar{S}}$ is the total delay of the steps not in $S$. This can be significantly tighter for some delay values. For simplicity of presentation, we present all bounds in the paper in the worst-case form $\sqrt{D}$, and note that our bound can also attain the tighter delay dependence in the adversarial case (see Corollary 5.4).*

While the regret bounds of delayed Multi-armed Bandit under stochastic losses and under adversarial losses are well understood separately, the following question remains open:

> *Is there a single algorithm that, without knowing the nature of the losses a-priori in delayed environment, can achieve the optimal regret bounds in both regimes simultaneously?*

Such an algorithm is often referred to as a *best-of-both-worlds* algorithm. Masoudian et al. [21, 22] have made significant progress toward answering this question. Their regret bound is $O(\sqrt{TK} + \sqrt{D} + K\sigma_{max} + \Phi^*)$ in the adversarial regime and $O\left(\sum_{i:\Delta_i>0}\left(\frac{\log(T)}{\Delta_i} + \frac{\sigma_{max}}{\log(K)\Delta_i}\right) + \Phi^*\right)$ in the stochastic regime, where $\Phi^* = \min\{d_{max}K^{2/3}, \sqrt{DK^{2/3}}\}$. However, these bounds are still not optimal.[1]

**Our contributions.** In this work, we affirmatively answer the above question and present a new best-of-both-worlds algorithm for Multi-armed Bandits (MAB) with delayed feedback that simultaneously achieves the near-optimal regret bounds under both stochastic and adversarial losses. Specifically:

- In the adversarial regime our algorithm guarantees optimal $\tilde{O}(\sqrt{TK} + \sqrt{D})$ regret.
- In the stochastic regime our algorithm guarantees optimal $O(\sum_{i \neq i^\star}(\frac{\log(T)}{\Delta_i} + \frac{1}{K}\sigma_{max}\Delta_i))$ regret.

In the adversarial regime, compared to Masoudian et al. [22] we have an extra logarithmic factor in the $\sqrt{TK}$ term, which is independent of the delay. However, we eliminate the additive $\Phi^*$ in their

---

[1] An additional benefit of Masoudian et al. [21, 22] is that they are *any time* algorithms, that do no need to know the horizon $T$ in advance.

[2] In these papers the $\sqrt{D}$ is actually $\min_{S \subseteq [T]}\left\{|S| + \sqrt{D_{\bar{S}}}\right\}$, where $D_{\bar{S}}$ is the total delay of the steps not in $S$. We wrote the worst-case for the simplicity of the table.

bound, which is significant when $d_{max}$ is very large; even a single large delay causes the regret to scale as $\sqrt{DK^{2/3}}$ rather than our $\sqrt{D}$ delay term, which is tight to the lower bound of Masoudian et al. [21].

Even more significantly, in the stochastic regime, our bound improves the $O(\sum_i \frac{\sigma_{max}}{\Delta_i \log(K)} + \Phi^*)$ term from the bound of [22] to $O(\frac{1}{K} \sum_i \sigma_{max} \Delta_i)$. That is, for each term in the summation, we achieve an improvement by a factor of $\frac{K}{\Delta_i^2 \log(K)}$. This is a significant improvement. For example, consider the simple case of fixed delay $d$, which implies $\sigma_{max} = d$, and constant number of actions. For any sub-optimality gaps our regret is at most $\sqrt{T} + d$ while there is a setting where the regret of [22] is at least $\sqrt{dT}$. Moreover, if the maximum delay is large, $\Phi^*$ can be as large as $\sqrt{D}$, offering no improvement over the additive delay term in the adversarial setting.

Our bound in the stochastic regime represents an improvement even compared to state-of-the-art results for algorithms specifically designed for the stochastic case. Specifically, Joulani et al. [18] provides the best-known result for stochastic losses with adversarial delays where their bound includes an additive term of $\sum_{i \neq i^*} \sigma_{max} \Delta_i$, which we improve by a factor of $\Theta(K)$. While Lancewicki et al. [19] reduce this dependence on $K$, their result applies only to the case of stochastic delays. Moreover, their regret bound scales with the maximal sub-optimality gap, rather than the average. For example, in the simple case of a fixed delay $d$, their additive term is $d \max_i \Delta_i$, whereas ours is $\frac{d}{K} \sum_i \Delta_i$, offering a strictly better dependence on the problem parameters in many scenarios.

## 1.1 Additional Related work

**Delayed MAB with stochastic losses.** The problem was first addressed by Dudik et al. [9], who analyzed the case of constant delays and established a regret bound with linear dependence on the delay. This line of work was extended by Joulani et al. [18], who allowed the delays to change through time. Subsequent work introduced several important refinements: Zhou et al. [38] distinguished between arm-dependent and arm-independent delays; Pike-Burke et al. [23] introduced an aggregated rewards model where only the sum of rewards that arrive at the same round is observed; and Lancewicki et al. [19] studied delays in the contexts of reward-dependent or reward-independent delays. More recently, Tang et al. [29], Schlisselberg et al. [25] and Zhang et al. [37] studied settings in which the delay is equal to the payoff.

**Delayed MAB with adversarial losses.** Delayed feedback have also been explored in adversarial settings, where both rewards and delays can be chosen adversarially. Quanrud and Khashabi [24] studied this problem in the full-information setting. The bandit setting was first addressed by Cesa-Bianchi et al. [6], who analyzed the case of constant delay. This line of work was extended by Thune et al. [30] who considered general adversarial delays under the assumption that the delay is known at the time the arm is pulled.[3] Subsequently, Gyorgy and Joulani [13] and Zimmert and Seldin [40] removed this assumption and analyzed the case where the delay is unknown at the time of the action. Finally, Van Der Hoeven and Cesa-Bianchi [32] extended the setting to allow for arm-dependent delays.

**"Best of Both Worlds" without delays.** The "Best of Both Worlds" framework in multi-armed bandits was introduced by Bubeck et al. [3], who proposed an algorithm that initially follows a stochastic-style strategy but switches to a standard adversarial algorithm upon detecting signs of adversarial losses. This adaptive approach was further developed by Auer and Chiang [1]. An alternative perspective is to start with an adversarial-style algorithm and prove that it achieves instance-dependent regret bounds in stochastic settings as well. In this direction, Seldin and Slivkins [27] and Seldin and Lugosi [26] adapted the EXP3 algorithm to perform well in both regimes, while Zimmert and Seldin [39], Dann et al. [8], Ito et al. [17] extended this idea to Follow-The-Regularized-Leader (FTRL), achieving optimal performance across both adversarial and stochastic settings.

---

[3]A similar result appeared in Bistritz et al. [2], however, there are some issues in their analysis - for further details see footnote 1 in Gyorgy and Joulani [13].

---
**Protocol 1** Delayed MAB
---
1: **for** $t \in [T]$ **do**
2:     Agent picks an action $a_t \in [K]$.
3:     Agent incurs loss $\ell_t(a_t)$ and observes feedback $\{(\ell_s(a_s), d_s) : t = s + d_s\}$.
---

## 2 Settings

We study the Multi-armed Bandit (MAB) problem with delayed feedback, summarized in Protocol 1. In each round $t = 1, 2, \ldots, T$, an agent chooses an arm $a_t \in [K]$ and suffers loss $\ell_t(a_t)$, where $\ell_t(\cdot) \in [0, 1]^K$ can be either stochastic or adversarial. Under the stochastic regime for each $i \in [K]$, $\{\ell_t(i)\}_{t=1}^T \overset{i.i.d}{\sim} \mathcal{D}_i$ where $\mathcal{D}_i$ is some distribution with expectation $\mu_i$. Under the adversarial regime the loss sequence $\{\ell_t\}_{t=1}^T$ are chosen arbitrarily by an oblivious adversary. Unlike the standard MAB setting, the agent does not immediately observe $\ell_t(a_t)$ at the end of round $t$; rather, only after $d_t$ rounds (namely, at the end of round $t + d_t$) the tuple $(t, \ell_t(a_t))$ is received as feedback. The delays $\{d_t\}_{t=1}^T$ are chosen by an oblivious adversary and are unknown at action time.

The performance of the agent is measured as usual by the the difference between the algorithm's cumulative expected loss and the best possible total expected reward of any fixed arm:

$$\mathcal{R}_T = \mathbb{E}\left[\sum_{t=1}^T \ell_t(a_t)\right] - \min_i \mathbb{E}\left[\sum_{t=1}^T \ell_t(i)\right].$$

In the stochastic case the regret can also be written as,

$$\mathcal{R}_T = \mathbb{E}\left[\sum_{t=1}^T \mu_{a_t}\right] - T\mu_{i^*} = \mathbb{E}\left[\sum_{t=1}^T \Delta_{a_t}\right],$$

where $i^*$ denotes the optimal arm and $\Delta_i = \mu_i - \mu_{i^*}$ for all $i \in [K]$.

**Additional notation.** We denote the total delay by $D = \sum_{t=1}^T d_t$ and the maximal delay by $d_{max} = \max_{t \in [T]} d_t$. The amount of missing feedback at time $t$ is defined by $\sigma(t) = |\{\tau \mid \tau \leq t, \tau + d_\tau > t\}|$ and the maximum over $\sigma(t)$ is denoted by $\sigma_{max} = \max_{t \in [T]} \sigma(t)$. The rounds observed before and available at round $t$ are denoted by $B(t) = \{s : s + d_s < t\}$. For $X \in \mathbb{R}$, $[X]$ denotes the set of all positive integers $\leq X$.

**Notation for the algorithms.** Let $S$ denotes a sequence of rounds that the algorithm process. $S_{:n}$ is the first $n$ elements in $S$ and $S_{:-n}$ is $S$ except for the last $n$ elements. $n_i(S)$ is the number of pulls of arm $i$ in the rounds of $S$, $\hat{\mu}_i(S) = \frac{1}{n_i(S)} \sum_{s \in S : a_s = i} l_i(s)$ is the empirical mean over $S$ and $\text{width}_i(S) = \min\left\{1, \sqrt{\frac{2\log(T)}{n_i(S)}}\right\}$ is a confidence width. $\text{ucb}_i(S) = \min\{\hat{\mu}_i + \text{width}_i(S), \text{ucb}_i(S_{:-1})\}$ and $\text{lcb}_i(S) = \max\{\hat{\mu}_i - \text{width}_i(S), \text{lcb}_i(S_{:-1})\}$ are upper and lower confidence bounds with respect to the empirical average. The algorithm also maintains confidence bounds around an average importance sampling estimator. Let $\overline{L}_i(S) = \sum_{s \in S} \frac{\mathbb{1}[a_s = i]\ell_i(s)}{p_i(s)}$ be the sum of the estimators over rounds in $S$, and $\overline{\mu}_i(S) = \frac{1}{|S|}\overline{L}_i(S)$ be the average. We also define $\overline{\text{width}}(S) = \min\left\{1, \sqrt{\frac{2K\log(T)}{|S|}}\right\}$, $\overline{\text{lcb}}_i(S) = \max\{\overline{\mu}_i(S) - \overline{\text{width}}(S), \overline{\text{lcb}}_i(S_{:-1})\}$ and $\overline{\text{ucb}}_i(S) = \min\{\overline{\mu}_i(S) + \overline{\text{width}}(S), \overline{\text{ucb}}_i(S_{:-1})\}$. Finally, we define $\text{ucb}^*(S) = \min_i\{\text{ucb}_i(S), \overline{\text{ucb}}_i(S)\}$.

## 3 Algorithm

Our algorithm, sketched in Algorithm 2 and formally described in Algorithm 5, builds on the SAPO algorithm of Auer and Chiang [1]. The main idea is to integrate an external algorithm for adversarial settings, ALG. Our algorithm initially follows a stochastic-like strategy while monitoring whether the environment exhibits stochastic behavior. If this assumption is violated, it switches to ALG.

At its core, the algorithm is based on a successive elimination (SE) framework [11], maintaining a set of active arms played with equal probability. It tracks a confidence bound, width, which defines upper and lower estimates for each arm's mean. When an arm is found to be non-optimal, it is eliminated. However, unlike standard SE methods, the algorithm continues to play eliminated arms but with reduced probability. This accounts for the possibility that losses are adversarial—an arm that appears suboptimal at one point may later turn out to be optimal. To verify the stochastic nature of arms, the algorithm employs the BSC procedure to assess the nature of active arms, and a more advanced procedure EAP for assessing and determining the sampling probability of non-active arms.

---

**Algorithm 2** Sketch of Delayed SAPO Algorithm

---

**Require:** Number of arms $K$, number of rounds $T \geq K$, Algorithm ALG.

1: Initialize active arms $\mathcal{A} = \{1, \ldots, K\}$, $S = \langle \rangle$
2: **for** $t = 1, 2, \ldots, T$ **do**
3:     **for** $s \in B(t) \setminus S$ **do**                 ▷ *Iterating newly received feedback*
4:         $S = S + \langle s \rangle$
5:         **if** not $\mathtt{BSC}(S)$ **then**        ▷ *Non-stochastic behavior on active arms (Procedure 3)*
6:            Switch to ALG.
7:         $\mathcal{A} = \mathcal{A} \setminus \{i \in \mathcal{A} : \hat{\mu}_i(S) - 9\,\mathrm{width}_i(S) > \mathrm{ucb}^*(S)\}$       ▷ *Elimination*
8:         **for** Each eliminated arm $i$ **do**
9:            $E_i = 0, r_i = 1, C_i^{p_i^1 \cdot 2^{-j}} = \emptyset \; \forall j \in [\log(T)]$
10:    **for** $i \in ([K] \setminus \mathcal{A})$ **do**
11:       $p_i(t), err = \mathtt{EAP}(i)$ ▷ *Get the reduced probability for the non-active arm (Procedure 4)*
12:       **if** $err$ **then**              ▷ *Non-stochastic behavior on nonactive arms*
13:          Switch to ALG.
14:    $\forall i \in \mathcal{A} \quad p_i(t) = \left(1 - \sum_{j \in ([K] \setminus \mathcal{A}(t))} p_j(t)\right) / |\mathcal{A}(t)|$    ▷ *Equal probability for active arms*
15:    Sample $a_t \sim p(t)$, observe feedback and update variables

---

**Basic Stochastic Checks (BSC) Subroutine.** This procedure performs two checks. The first ensures that an unbiased estimate of the mean of each arm remains within its confidence interval, expanded by an additional radius. In the stochastic regime, using standard concentration bounds we have that with high probability,

$$\overline{\mathrm{lcb}}_i(S) \leq \mu_i \leq \overline{\mu}_i(S) + \overline{\mathrm{width}}(S); \qquad \overline{\mu}_i(S) - \overline{\mathrm{width}}(S) \leq \mu_i \leq \overline{\mathrm{ucb}}_i(S).$$

Thus, in line 1 of Procedure 3, we check that the above conditions are met.

The second check in BSC constructs a lower bound on the regret and verifies that it is indeed smaller than the expected regret in the stochastic regime, which can be shown to be $\tilde{O}(\sqrt{TK} + \sigma_{max})$ under stochastic losses. To define this lower bound, we use the fact that, with high probability, $\mu^* \leq \mathrm{ucb}^*(S)$. Thus, $\sum_{s' \in S} \left( l_{a_{s'}}(s') - \mathrm{ucb}^*(S) \right)$ is a lower bound on the regret, which forms the condition in line 3 of the procedure.

---

**Procedure 3** Basic Stochastic Checks (BSC) Subroutine

---

**Require:** Series of processed pulls $S$

1: **if** $\exists i \in \mathcal{A} : \overline{\mu}_i(S) \notin [\overline{\mathrm{lcb}}_i(S) - \overline{\mathrm{width}}(S), \overline{\mathrm{ucb}}_i(S) + \overline{\mathrm{width}}(S)]$ **then**
2:    **return** False
3: **if** $\sum_{s' \in S} \left( l_{a_{s'}}(s') - \mathrm{ucb}^*(S) \right) > 272\sqrt{KT \log(T)} + 10\sigma_{max}(t) \log(K)$ **then**
4:    **return** False
   **return** True

---

**Eliminated Arms Processing (EAP) Subroutine.** Since we do not know in advance whether we are in the stochastic or adversarial regime, we cannot completely eliminate an arm — if we did, the adversary could assign losses of $0$ after elimination of an arm, and we would never detect this. Therefore, we maintain a positive sampling probability even for eliminated actions. EAP maintains these probabilities for eliminated arms and checks whether the estimated loss is significantly smaller than the empirical mean at elimination. Intuitively, if we are in the stochastic regime, we want the probability of playing an eliminated arm to decrease over time. Conversely, if we suspect the loss after elimination is significantly smaller than the empirical mean at the elimination time, we increase

**Procedure 4** Sketch of Eliminated Arms Processing (EAP) Subroutine

**Note:** The variables $E_i$, $r_i$, and $C_i^p$ are initialized in Algorithm 2 and updated through multiple calls of this procedure.

**Require:** Arm $i$

1:   $p := p_i^{r_i}$, $\tilde{\mu}$ is the empirical average at the elimination time of $i$
2:   Let $B_i^p$ be observed rounds after elimination in which $i$ was played and the sampling probability was $p$
3:   **while** $B_i^p \setminus C_i^p \neq \emptyset$ **do**
4:     **for** $s \in B_i^p \setminus C_i^p$ **do**
5:       $C_i^p = C_i^p \cup \{s\}$
6:       Let $S_i^{r_i}$ be the samples processes so far in phase $r_i$
7:       **if** $|S_i^{r_i}|\tilde{\mu}_i - \bar{L}(S_i^{r_i}) \geq \frac{1}{4}\tilde{\Delta}_i N_i^{r_i}$ **then**         $\triangleright$ *phase error*
8:         $E_i = E_i + 1$, $N_i^{r_i+1} = \max\left\{N_i^1, \frac{1}{2}N_i^{r_i}\right\}$, $p_i^{r_i+1} = \min\left\{p_i^1, 2p_i^{r_i}\right\}$, $r_i = r_i + 1$
9:         **if** $E_i \geq 3\log(T)$ **then return** $0$, True      $\triangleright$ *Switch to adversarial algorithm*
10:        **break**
11:       **if** $|S_i^{r_i}| = \lfloor N_i^{r_i} \rfloor$ **then**         $\triangleright$ *phase ended*
12:         $N_i^{r_i+1} = 2N_i^{r_i}$, $p_i^{r_i+1} = \frac{1}{2}p_i^{r_i}$, $r_i = r_i + 1$
13:        **break**
14:     $p := p_i^{r_i}$
15:   **return** $p_i^{r_i}$, False

---

that arm's probability to monitor it more closely. If there is sufficient confidence that the arm does not behave stochastically, we switch to the adversarial algorithm.

In more detail, the probability of playing an eliminated arm $i$ is updated in discrete phases. Let $\tilde{S}_i$ be the set of processed rounds at the time of elimination of arm $i$. We denote $\tilde{\Delta}_i = 8\text{width}_i(\tilde{S}_i)$, i.e the width at elimination time. As we'll later see in the analysis, $\tilde{\Delta}_i$ is indeed a good estimate of the sub-optimality gap of arm $i$ in the stochastic case (see Lemma D.10). Each phase $r$ has a maximum length $N_i^r = \Theta(1/(p_i^r \tilde{\Delta}_i^2))$, where $p_i^r$ is the sample probability of arm $i$ in its $r$th phase and $p_i^1 = \frac{1}{2K} + \frac{n_i(\tilde{S}_i)}{2T}$.[4] This value is always $\Omega(1/K)$, but can be as high as a uniform probability over the active arms at the time of elimination. If we reach the maximum length $N_i^r$, then we have acquired additional $N_i^r p_i^r = \Theta(1/\tilde{\Delta}_i^2)$ samples from arm $i$. In this case, we can safely halve the sampling probability of arm $i$ and start a new phase with a doubled maximum length (line 11). During the phase, we monitor whether the average importance sampling estimate of the loss $\bar{\mu}(S_i^r)$ is smaller than $\tilde{\mu}_i = \hat{\mu}_i(\tilde{S}_i)$ by more than $\Theta(\tilde{\Delta}_i N_i^r/|S_i^r|)$, where $S_i^r$ is the sequence of processed rounds in phase $r$. If this condition is met, referred to as a "phase error", in means that the observed losses appear slightly non-stochastic. Thus, we terminate the phase but now double the sampling probability of arm $i$ and halve the maximum phase length accordingly (line 6).

In the stochastic regime, phase errors occur with a constant probability, but the probability that they will happen $\Theta(\log(T))$ times is negligible. In such cases, we transition to the adversarial algorithm.

During a phase with sampling probability $p$, we process only the observed rounds after elimination in which arm $i$ was played and the sampling probability was $p$. If a sample is observed with a different sampling probability $p'$, it is stored in a *"probability bank"* which we denote by $B_i^{p'}$ and is processed only if a new phase is initiated with probability $p'$. The probability banks allow us to utilize most samples, even if they are observed after their respective phases end, and play an important role in removing a factor of the number of phases ($\Theta(\log(T))$) from the delay term in the regret.

---

[4]We note that the initial probability assigned in the first phase differs from that in Auer and Chiang [1], and is crucial for obtaining adversarial regret bound that scales with $\sqrt{KT}$ instead of $K\sqrt{T}$ achieved in [1].

# 4 Stochastic analysis

**Theorem 4.1** *The regret in the stochastic settings is bounded by:*

$$\mathcal{R}_{sto} \leq O\left(\sum_{i=1}^{K} \frac{\log(T)}{\Delta_i} + \sigma_{max}\Delta_{avg}\log(K)\right)$$

The first term above is the optimal MAB regret under stochastic losses without delays. The second term is the additional regret due to delay and, in general, cannot be improved—except for the $\log(K)$ factor, due to the lower bound for constant delays (see Table 1). We note that with a more involved algorithm and analysis, we are able to eliminate the $\log(K)$ factor and match this lower bound. For simplicity of presentation, the full details are deferred to Appendix F. The dependence on $\sigma_{max}$ improves upon the BoBW result of Masoudian et al. [22] in the stochastic regime by a factor of $\tilde{O}(K/\Delta_i^2)$ for each $i$. Moreover, it is tighter by a factor of $K$ compared to the best previous known algorithm that specifically designed for this regime (Joulani et al. [18]).

**Proof sketch:** The total regret can be decomposed as,

$$\mathcal{R}_{sto} = \sum_{i\in[K]} m_i(T)\Delta_i = \sum_{i\in[K]} m_i(\tau_i)\Delta_i + \sum_{i\in[K]} (m_i(T) - m_i(\tau_i))\Delta_i, \tag{1}$$

where $m_i(t)$ is the number of pulls of arm $i$, up to time $t$ and $\tau_i$ is the elimination time of arm $i$. The first term above is the regret up to elimination and second term is the regret after elimination. (Recall that we need to keep sampling eliminated arms.)

**Regret up to elimination.** The regret before elimination analysis largely follows standard Stochastic Elimination (SE) with delayed feedback arguments. However, achieving dependence on $\Delta_{avg}$ rather than $\Delta_{max}$ necessitates a new algorithmic component and technical argument. We start by further decomposing the regret up to elimination:

$$\sum_{i\in[K]} m_i(\tau_i)\Delta_i = \sum_{i\in[K]} n_i(\tau_i)\Delta_i + \sum_{i\in[K]} (m_i(\tau_i) - n_i(\tau_i))\Delta_i,$$

where $n_i(t)$ is the number of *observed* samples from arm $i$. Similar to standard non-delayed SE analysis, we can show that with high probability, each suboptimal arm is eliminated whenever $\Theta\left(\frac{\log(T)}{\Delta_i^2}\right)$ samples from arm $i$ have been observed. Thus, $n_i(\tau_i)\Delta_i = \Theta\left(\frac{\log(T)}{\Delta_i}\right)$. For the second term above, recall that the number of missing feedback is bounded by $\sigma_{max}$; but only a fraction of the missing feedback is from arm $i$. Loosely speaking, if $p_i^{max} = \max_{t\leq\tau_i} p_i(t)$ is the maximal probability of sampling $i$ before elimination, then the number of missing feedback from arm $i$ at time $\tau_i$ is roughly bounded by $m_i(\tau_i) - n_i(\tau_i) \leq \sigma_{max}p_i^{max}$. Further note that if $\kappa_i$ is the number of active arms at the time of elimination then $p_i^{max} \leq \frac{1}{\kappa_i}$. Overall, the total regret up to elimination is bounded by

$$O\left(\sum_i \frac{\log(T)}{\Delta_i} + \sum_i \frac{\sigma_{max}}{\kappa_i}\Delta_i\right)$$

For the second term, each $\Delta_i$ can be trivially bounded by $\Delta_{max}$, and $\sum_i 1/\kappa_i \leq \sum_i 1/i \leq 1 + \log(K)$, resulting in $\sum_i \frac{\sigma_{max}}{\kappa_i}\Delta_i \leq O(\sigma_{max}\Delta_{max}\log(K))$. In order to have dependency with respect $\Delta_{avg}$ instead of $\Delta_{max}$ a more detailed argument is required. Unlike regular SE algorithms, an arm isn't eliminated when the ucb of some other arm is lower than its lcb. Instead, the algorithm eliminates when there are multiple widths between the two (see line 7 in algorithm 2). This stricter condition ensures that arms are roughly eliminated in decreasing order of $\Delta_i$. Specifically, we show the following lemma:

**Lemma 4.2** *If arm $i_1$ was eliminated before $i_2$ then, $\Delta_{i_2} \leq 20\Delta_{i_1}$.*

We note that this is relatively general trick that may be used in other regimes; see remark 2.

For the first half of eliminated arms where $\kappa_i \geq K/2$, the additive delay term is at most order of $\sum_{i:\kappa_i>K/2} \frac{\sigma_{max}}{\kappa_i}\Delta_i \leq \sigma_{max}\Delta_{avg}$. Using the above lemma we show that for second half of

eliminated arms $\Delta_i = O(\Delta_{avg})$, yielding an additive delay term of at most $O(\sigma_{\max}\Delta_{avg}\log(K))$. Overall we get that the regret up to elimination is bounded by $\sum_{i\in[K]} m_i(\tau_i)\Delta_i \lesssim \sum_i \frac{\log(T)}{\Delta_i} + \sigma_{\max}\Delta_{avg}\log(K)$.

**Regret after elimination.** For the regret after elimination, we break the number of pulls of arm $i$ after elimination for pulls that where processed by algorithm and pulls that where not processed by the algorithm (either because the feedback had not returned or the samples remained in the probability bank):

$$m_i(T) - m_i(\tau_i) = \sum_{r=1}^{r_i} n_i(S_i^r) + \sum_{j=0}^{\log(T)} n_i(M_i^{p_i^1 2^{-j}}), \tag{2}$$

where $r_i$ is the total number of phases of arm $i$, $S_i^r$ are the samples processes at phase $r$ and $M_i^p$ denotes the post-elimination rounds where the probability of pulling arm $i$ was $p$, but these were not processed by the algorithm (either because the feedback was not observed or the rounds remained unprocessed in the probability bank).

Recall that the maximum length of phase $r$ is $N_i^r = \Theta(1/(p_i^r \tilde\Delta_i^2))$. Additionally, the fact that arms are only eliminated when the empirical average exceeds ucb* by more than multiple `widths` allows us to show that $\tilde\Delta_i \approx \Delta_i$ (see Lemma D.10). Using standard concentration bounds, $n_i(S_i^r) \approx N_i^r p_i^r \approx 1/\Delta_i^2$. To bound the number of phases, note that the maximum phase length can be either doubled or halved. The number of times it is halved in the stochastic regime is at most $3\log(T)$ with high probability (see Lemmas D.5 and D.19), where in case of a failure event, we switch to the adversarial algorithm. Since the number of times it is halved is bounded by $O(\log(T))$, we can also bound the number of times it is doubled before reaching the time horizon $T$. Formally, in Lemma C.4, we bound the total number of phases by $7\log(T)$. Therefore the first term in Equation (2) is bounded by $O(\frac{\log T}{\Delta_i^2})$ and the regret from these rounds is $O(\frac{\log T}{\Delta_i})$.

For the second term of eq. (2), note that the size of $M_i^{p_i^1 2^{-j}}$ is at most $\sigma_{max}$, but only a small fraction of those rounds belongs to arm $i$. Since the probability of pulling arm $i$ in these rounds was $p_i^1 2^{-j}$ we have that $n_i(M_i^{p_i^1 2^{-j}}) \approx \sigma_{max} p_i^1 2^{-j}$. Summing over this geometric series gives us $\sum_{j=0}^{\log(T)} n_i(M_i^{p_i^1 2^{-j}}) = O(\sigma_{max} p_i^1)$. Recall that $p_i^1 = \frac{1}{2K} + \frac{n_i(\tau_i)}{2T}$. Since the probability of pulling arm $i$ before elimination is at most $1/\kappa_i$, where $\kappa_i$ is the number of active arms at the time of elimination, $n_i(\tau_i)/T \le n_i(\tau_i)/\tau_i \lesssim 1/\kappa_i$. That is, $p_i^1 \le O(1/\kappa_i)$. We get that the total regret after elimination from unprocessed pulls (multiplying the second term in eq. (2) by $\Delta_i$ and summing over $i$) is of order $\sum_i \frac{\sigma_{max}}{\kappa_i}\Delta_i$. Again, leveraging the fact that arms are eliminated roughly in decreasing order of their sub-optimality gaps, we can bound the last sum by $\sigma_{max}\Delta_{avg}\log(K)$. ∎

**Remark 2** *As mentioned in the proof sketch, our algorithm adds additional* `width` *to the elimination inequality, which makes the eliminated arms to be in descending order of their sub-optimality gap. We stress that this is a general trick that can be applied in any SE-based algorithm. Specifically, for every SE-based algorithm for delayed feedback (e.g [20, 25]), this will make their additive term be dependent on $\Delta_{avg}$ instead of $\Delta_{max}$.*

## 5   Adversarial Analysis

**Theorem 5.1** *Assume that ALG has a regret guarantee of $R_{ALG}$ in terms of $T$, $K$, $D$ and possibly $d_{max}$ and $\sigma_{max}$. Then, the regret in the adversarial setting is bounded by:*

$$R_{adv} \le O\left(\sqrt{KT\log(T)} + \log(K)\sigma_{max} + R_{ALG}\right)$$

The $\log(K)$ factor can be removed with a slight algorithm modification. We deferred the details to Appendix F to reduce the complexity of the already intricate main algorithm.

**Proof sketch:** Fix action $i \in [K]$. Let $\bar T$ be the time the algorithm switches to `ALG`. Clearly, the regret after the switch is bounded by $R_{ALG}$, so we focus on the regret up to time $\bar T$. First, we

decompose it to the following three terms:

$$\underbrace{\mathbb{E}\left[\sum_{t=1}^{\bar{T}}\left[l_{a_t}(t)-\text{ucb}^*(\bar{S})\right]\right]}_{(3)}+\underbrace{\mathbb{E}\left[\sum_{t=1}^{\tau_i-1}\left[\text{ucb}^*(\bar{S})-l_i(t)\right]\right]}_{(4)}+\underbrace{\mathbb{E}\left[\sum_{t=\tau_i}^{\bar{T}}\left[\text{ucb}^*(\bar{S})-l_i(t)\right]\right]}_{(5)}$$

where $\bar{S}$ is the value of $S$ when the algorithm switches to ALG and $\tau_i$ is the elimination time of arm $i$. Term (3) is bounded by $O(\sqrt{KT\log(T)}+\log(K)\sigma_{max})$ due to the second check of BSC. To bound term (4), note that $\bar{L}_i$ is an unbiased estimator of $L_i$. By Wald's equation, $\mathbb{E}\left[L_i(\tilde{S}_i)\right]=\mathbb{E}\left[\bar{L}_i(\tilde{S}_i)\right]=\mathbb{E}\left[|\tilde{S}_i|\bar{\mu}_i(\tilde{S}_i)\right]$, where $\tilde{S}_i$ is the set of rounds in which arm $i$ was observed before elimination. Now, since we have not switched to ALG yet, by the first check of BSC we know that for every realization $S$ of $\tilde{S}_i$,

$$\bar{\mu}_i(S)\geq\overline{\text{lcb}}_i(S)-\overline{\text{width}}(S)\geq\overline{\text{ucb}}_i(S)-3\overline{\text{width}}(S)\geq\text{ucb}^*(S)-3\overline{\text{width}}(S),$$

where in the second inequality we used the fact that $\overline{\text{ucb}}_i(S)-\overline{\text{lcb}}_i(S)\leq2\overline{\text{width}}(S)$ for any $S$ and the last inequality is by definition of $\text{ucb}^*$. Multiplying both sides above by $|\tilde{S}_i|$ gives us

$$\mathbb{E}\left[\sum_{t\in\tilde{S}_i}l_i(t)\right]\geq\mathbb{E}\left[|\tilde{S}_i|\text{ucb}^*(\tilde{S}_i)-3\sqrt{2|\tilde{S}_i|K\log(T)}\right]\geq\mathbb{E}\left[|\tilde{S}_i|\text{ucb}^*(\bar{S})-3\sqrt{2TK\log(T)}\right]$$

Rearranging the terms above we get that $\mathbb{E}\left[\sum_{t\in\tilde{S}_i}\left[\text{ucb}^*(\bar{S})-l_i(t)\right]\right]\leq3\sqrt{2TK\log(T)}$. Hence,

$$(4)=\mathbb{E}\left[\sum_{t\in\tilde{S}_i}\left[\text{ucb}^*(\bar{S})-l_i(t)\right]\right]+\mathbb{E}\left[\sum_{t\in[\tau_i\backslash\tilde{S}_i]}\left[\text{ucb}^*(\bar{S})-l_i(t)\right]\right]\leq3\sqrt{2TK\log(T)}+\sigma_{max}$$

where we've used the fact that ucb decreases over time and $|[\tau_i-1]\backslash\tilde{S}_i|\leq\sigma_{max}$.

The core difficulty in the adversarial analysis is bounding (5) — ensuring that an eliminated arm doesn't become much better than the active arms, before switching to ALG. Let us further decompose (5) to the phases of arm $i$:

$$\mathbb{E}\left[\sum_{t=\tau_i}^{\bar{T}}\left[\text{ucb}^*(\bar{S})-l_i(t)\right]\right]=\mathbb{E}\left[\sum_{r=1}^{r_i(T)}\sum_{t\in S_i^r}\left[\text{ucb}^*(\bar{S})-l_i(t)\right]\right]=\mathbb{E}\left[\sum_{r=1}^{r_i(T)}\left(|S_i^r|\text{ucb}^*(\bar{S})-L_i(S_i^r)\right)\right]$$

The main tool to upper bound the optimality of an eliminated arm is the check in Line 9 of EAP. This checks that the estimated loss (using an importance sampling estimator) isn't much higher than the loss observed when the arm was active. Using the condition in Line 9 of EAP and by bounding the difference between the loss estimator of the phase and the actual cost in terms of $N_i^r$ and $\tilde{\Delta}_i$ we show the following lemma (the proof is deferred to the appendix - see Lemma E.2):

**Lemma 5.2** *For every arm $i$ and phase $r$ we have:*

$$\mathbb{E}_i^r\left[|S_i^r|\text{ucb}^*(\tilde{S}_i)-L_i(S_i^r)\right]\leq\frac{3}{8}\tilde{\Delta}_iN_i^r-\frac{9}{8}\tilde{\Delta}_i\mathbb{E}_i^r[|S_i^r|],$$

*where $\mathbb{E}_i^r$ is the expectation conditioned on the observed history by the beginning of the $r$th phase of arm $i$.*

Note that the difference between $\text{ucb}^*$ and the expected loss depends on the relationship between $|S_i^r|$ and $N_i^r$. If the phase finished successfully ($|S_i^r|=N_i^r$), the expected loss exceeds $\text{ucb}^*$. If the phase was erroneous, then we may have $|S_i^r|\ll N_i^r$, and the expected loss can be better than $\text{ucb}^*$. Specifically:

$$\text{Finished phase:}\quad|S_i^r|\text{ucb}^*(\bar{S})-\mathbb{E}[L_i(S_i^r)]\leq-\frac{3}{4}\tilde{\Delta}_iN_i^r$$

$$\text{Erroneous phase:}\quad|S_i^r|\text{ucb}^*(\bar{S})-\mathbb{E}[L_i(S_i^r)]\leq\frac{3}{8}\tilde{\Delta}_iN_i^r$$

The trick for bounding the sum of these bounds over the phases is that every successfully finished phase compensate for the erroneous phases after it, since the coefficient of $\tilde{\Delta}_i N_i^r$ under finished phases is twice as the coefficient for error phases. Since the algorithm halves $N_i^r$ after an error, even $O(\log(T))$ erroneous phases are eventually covered by the last successful phase. The precise argument is by induction and is rather technical. For full details, see the appendix in Lemma E.3. ∎

To conclude the proof we use the following lemma (proof is deferred to the appendix):

**Lemma 5.3** $\sigma_{max} \leq O\big(\min_{S \in [T]} \big\{|S| + \sqrt{D_{\bar{S}}}\big\}\big).$

**Corollary 5.4** *Using* ALG *as the algorithm from Zimmert and Seldin [40], we have:*

$$R_{adv} \leq \tilde{O}\bigg( \sqrt{KT \log(T)} + \min_{S \in [T]} \big\{|S| + \sqrt{D_{\bar{S}}}\big\} \bigg)$$

The first term in the regret bound is nearly optimal—up to the $\log(T)$ factor, it matches the worst-case regret for MAB without delays. The second term, which accounts for the delays, is also tight in general due to the lower bound of Masoudian et al. [21]. Our bound eliminates altogether the $\Phi^*$ term that appears in the BoBW bound of Masoudian et al. [22] for the adversarial regime (see Table 1). This is especially significant whenever $d_{\max}$ is very large (i.e., even if a single delay is large), in which case $\Phi^* = \sqrt{DK^{2/3}}$ while our delay term only scales with $\sqrt{D}$ in the worst case.

# 6 Discussion

We presented a novel algorithm for the delayed-BoBW problem that achieves a near-optimal regret bound simultaneously for both stochastic and adversarial losses. Additionally, our bounds in the stochastic regime improve even compared to algorithms specifically designed for the stochastic case. As mentioned, our algorithm follows the "adaptive approach" for BoBW—it begins with an algorithm that achieves optimal bounds in the stochastic setting, and upon identifying non-stochastic losses, it switches to an optimal algorithm for the adversarial setting. The alternative perspective Masoudian et al. [21, 22] offers a simpler algorithm but results in weaker bounds. It remains an open question whether algorithms of the latter type can achieve optimality in the delayed scenario. Apart from the simplicity of these type of algorithms, we also note that they typically have any-time guarantee, whereas ours either requires knowing T in advance or incurs an additional logarithmic factor. Additionally, while we considered worst-case adversarial delays, future research could explore delays with additional structure (such as i.i.d. or payoff-dependent delays), potentially yielding improved regret bounds.

# Acknowledgments

OS, TL and YM are supported by the European Research Council (ERC) under the European Union's Horizon 2020 research and innovation program (grant agreement No. 882396), by the Israel Science Foundation and the Yandex Initiative for Machine Learning at Tel Aviv University and by a grant from the Tel Aviv University Center for AI and Data Science (TAD).

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

# A  Notations

For a series $S$, we denote $S_{:k}$ to be the first $k$ elements of it. Additionally, $S_{:-1}$ is the series without the last element.

Additionally, for every variable defined inside the algorithm, in the analysis we will add $(t)$ to indicate that we refer to the value of this variable at the end of time $t$. For example, $r_i(t)$ is the value of $r_i$ at the end of time $t$.

| | | |
|---|---|---|
| $a_t$ | chosen arm at step $t$ | |
| $l_i(t)$ | loss of arm $i$ at step $t$ | |
| $d_t$ | delay at step $t$ | |
| $\mu_i$ | (stochastic) mean loss of arm $i$ | |
| $\bar{T}$ | switch to algorithm ALG point | |
| $n_i(S)$ | number of pulls of arm $i$ in $S$ | $\left| s \in S : a_s = i \right|$ |
| $m_i(t)$ | number of pulls of arm $i$ until time $t$ | $\left| s \in [t] : a_s = i \right|$ |
| $\bar{l}_i(t)$ | sample estimator for the loss of arm $i$ in step $t$ | $l_i(t)\frac{\mathbb{1}[a_t=i]}{p_i(t)}$ |
| $L_i(S)$ | total loss of arm $i$ in set $S$ | $\sum_{s \in S} l_i(s)$ |
| $\overline{L}_i(S)$ | sample estimator for the loss of arm $i$ w.r.t the steps in $S$ | $\sum_{s \in S} \bar{l}_i(s)$ |
| $\hat{\mu}_i(S)$ | average loss of arm $i$ w.r.t the steps in $S$ | $\frac{1}{n_i(S)}\sum_{s \in S: a_s=i} l_i(s)$ |
| $\overline{\mu}_i(S)$ | sample estimator average loss of arm $i$ w.r.t the steps in $S$ | $\frac{1}{|S|}\overline{L}_i(S)$ |
| $\text{width}_i(S)$ | average confidence width of arm $i$ w.r.t the steps in $S$ | $\min\left\{1, \sqrt{\frac{2\log(T)}{n_i(S)}}\right\}$ |
| $\overline{\text{width}}(S)$ | estimator confidence width w.r.t the steps in $S$ | $\min\left\{1, \sqrt{\frac{2K\log(T)}{|S|}}\right\}$ |
| $\text{lcb}_i(S)$ | average lower confidence bound of arm $i$ w.r.t set $S$ | $\max\left\{\hat{\mu}_i - \text{width}_i(S), \text{lcb}_i(S_{:-1})\right\}$ |
| $\text{ucb}_i(S)$ | average upper confidence bound of arm $i$ w.r.t set $S$ | $\min\left\{\hat{\mu}_i + \text{width}_i(S), \text{ucb}_i(S_{:-1})\right\}$ |
| $\overline{\text{lcb}}_i(S)$ | estimator lower confidence bound of arm $i$ w.r.t set $S$ | $\max\left\{\overline{\mu}_i(S) - \overline{\text{width}}(S), \overline{\text{lcb}}_i(S_{:-1})\right\}$ |
| $\overline{\text{ucb}}_i(S)$ | estimator upper confidence bound of arm $i$ w.r.t set $S$ | $\min\left\{\overline{\mu}_i(S) + \overline{\text{width}}(S), \overline{\text{ucb}}_i(S_{:-1})\right\}$ |
| $\text{ucb}^*(S)$ | | $\min_i\left\{\text{ucb}_i(S), \overline{\text{ucb}}_i(S)\right\}$ |
| $p_i^{max}(t_1, t_2)$ | maximum pull probability of arm $i$ in the interval | $\max_{t_1 \leq t \leq t_2} p_i(t)$ |
| $p_i^{min}(t_1, t_2)$ | minimum pull probability of arm $i$ in the interval | $\min_{t_1 \leq t \leq t_2} p_i(t)$ |
| $p_i^{max}(t)$ | | $p_i^{max}(0, t)$ |
| $p_i^{min}(t)$ | | $p_i^{min}(0, t)$ |
| $B(t)$ | Set of the steps whose feedback was observed | $\{s : s + d_s < t\}$ |
| $B_i^p(t)$ | Set of observed inactive steps that were pulled with prob. $p$ | $\{s \in B(t) : p_i(s) = p \wedge s \geq \tau_i\}$ |
| $M_i^p(t)$ | Set of inactive steps up to time $t$ that were pulled with prob. $p$ | $\{\tau_i \leq s \leq t : p_i(s) = p\}$ |

## B Algorithm

---

**Algorithm 5** Delayed SAPO Algorithm

---

**Require:** Number of arms $K$, number of rounds $T \geq K$, Algorithm ALG.

1: Initialize active arms $\mathcal{A} = \{1, \ldots, K\}$, $S = \langle \rangle$
2: **for** $t = 1, 2, \ldots, T$ **do**
3:     **for** $s \in B \setminus S$ **do**
4:        $S = S + \langle s \rangle$
5:        **if** not $\mathrm{BSC}(S)$ (Procedure 7) **then**
6:           Switch to ALG.
7:        $U(t) = \{i \in \mathcal{A} : \hat{\mu}_i(S) - 9\mathrm{width}_i(S) > \mathrm{ucb}^*(S)\}$          ▷ *Elimination*
8:        $\mathcal{A} = \mathcal{A} \setminus U$
9:        **for** $i \in U$ **do**          ▷ *Initialization for eliminated arms*
10:           Set $\tau_i = t$, $p_i^1 = \frac{1}{2K} + \frac{n_i(S)}{2T}$, $\tilde{S}_i = S$, $\tilde{\mu}_i = \hat{\mu}_i(S)$, $\tilde{\Delta}_i = 8\mathrm{width}_i(S)$, $N_i^1 :=$
              $1280/(p_i^1 \tilde{\Delta}_i^2)$, $E_i = 0$, $r_i = 1$, $S_i^1 = \langle \rangle$, $C_i^{p_i^1 \cdot 2^{-j}} = \emptyset \; \forall j \in [\log(T)]$
11:     **for** $i \in ([K] \setminus \mathcal{A})$ **do**
12:        $p_i(t), err = \mathrm{EAP}(i)$ (Procedure 6)
13:        **if** $err$ **then**
14:           Switch to ALG.
15:     $\forall i \in \mathcal{A} \quad p_i(t) = \left(1 - \sum_{j \in ([K] \setminus \mathcal{A}(t))} p_j(t)\right) / |\mathcal{A}(t)|$
16:     Observe feedback and update variables

---

**Procedure 6** Eliminated Arms Processing (EAP) Subroutine

---

**Require:** Arm $i$

1: $p := p_i^{r_i}$
2: **while** $B_i^p \setminus C_i^p \neq \emptyset$ **do**
3:     **for** $s \in B_i^p \setminus C_i^p$ **do**
4:        $S_i^{r_i} = S_i^{r_i} + \langle s \rangle$, $C_i^p = C_i^p \cup \{s\}$
5:        **if** $|S_i^{r_i}| \tilde{\mu}_i - \bar{L}(S_i^{r_i}) \geq \frac{1}{4} \tilde{\Delta}_i N_i^{r_i}$ **then**          ▷ *phase error*
6:           $E_i = E_i + 1$, $N_i^{r_i+1} = \max\left\{N_i^1, \frac{1}{2}N_i^{r_i}\right\}$, $p_i^{r_i+1} = \min\left\{p_i^1, 2p_i^{r_i}\right\}$, $S_i^{r_i+1} = \langle \rangle$,
          $r_i = r_i + 1$
7:           **if** $E_i \geq 3\log(T)$ **then return** $0$, True          ▷ *Switch to adversarial algorithm*
8:           **break**
9:        **if** $|S_i^{r_i}| = \lfloor N_i^{r_i} \rfloor$ **then**          ▷ *phase ended*
10:           $N_i^{r_i+1} = 2N_i^{r_i}$, $p_i^{r_i+1} = \frac{1}{2}p_i^{r_i}$ $S_i^{r_i+1} = \langle \rangle$, $r_i = r_i + 1$
11:           **break**
12:     $p := p_i^{r_i}$
       **return** $p_i^{r_i}$, False

---

**Procedure 7** Basic Stochastic Checks (BSC) Subroutine

---

**Require:** Series of processed pulls $S$

1: **if** $\exists i \in \mathcal{A} : \overline{\mu}_i(S) \notin [\overline{\mathrm{lcb}}_i(S) - \overline{\mathrm{width}}(S), \overline{\mathrm{ucb}}_i(S) + \overline{\mathrm{width}}(S)]$ **then**
2:     **return** False
3: **if** $\sum_{s' \in S} \left(l_{a_{s'}}(s') - \mathrm{ucb}^*(S)\right) > 272\sqrt{KT\log(T)} + 10\sigma_{max}(t)\log(K)$ **then**
4:     **return** False
    **return** True

---

## C General Lemmas

**Lemma C.1 (Freedman's Inequality, Theorem 1.1 in Tropp [31])** *Let $\{X_k\}_{k \geq 1}$ be a real valued martingale difference sequence adapted to a filtration $\{F_t\}_{t \geq 0}$. If $X_k \leq R$ a.s. Then, for all $t \geq 0$*

*and $\sigma^2 \geq 0$,*

$$\Pr\left[\exists k \geq 0 : \sum_{i=1}^{k} X_i \geq t \quad and \quad \sum_{i=1}^{k} \mathbb{E}\left[X_i^2 | F_{i-1}\right] \leq \sigma^2\right] \leq exp\left\{-\frac{t^2}{2\sigma^2 + 2Rt/3}\right\}$$

**Lemma C.2 (Lemma F.4 in Dann et al. [7])** *Let $\{X_t\}_{t=1}^{T}$ be a sequence of Bernoulli random and a filtration $\mathcal{F}_1 \subseteq \mathcal{F}_2 \subseteq ...\mathcal{F}_T$ with $\mathbb{P}(X_t = 1 \mid \mathcal{F}_t) = P_t$, $P_t$ is $\mathcal{F}_t$-measurable and $X_t$ is $\mathcal{F}_{t+1}$-measurable. Then, for all $t \in [T]$ simultaneously, with probability $1 - \delta$,*

$$\sum_{k=1}^{t} X_k \geq \frac{1}{2}\sum_{k=1}^{t} P_k - \log\left(\frac{1}{\delta}\right)$$

**Lemma C.3 (Consequence of Freedman's Inequality, e.g., Lemma 27 in Efroni et al. [10])** *Let $\{X_t\}_{t \geq 1}$ be a sequence of random variables, supported in $[0, R]$, and adapted to a filtration $\mathcal{F}_1 \subseteq \mathcal{F}_2 \subseteq ...\mathcal{F}_T$. For any $T$, with probability $1 - \delta$,*

$$\sum_{t=1}^{T} X_t \leq 2\sum_{t=1}^{T} \mathbb{E}[X_t \mid \mathcal{F}_t] + 4R\log\left(\frac{1}{\delta}\right).$$

**Lemma C.4** *For every arm $i$, $r_i(T) \leq 7\log(T)$*

**Proof:** Every phase must finish with Line 6 or Line 11. Let $r_1$ be the number of phases finished with Line 6. We have:

$$N_i^{\lceil 7\log(T)\rceil} \geq N_i^1 2^{-r_1} 2^{7\log(T)-r_1}$$
$$= N_i^1 2^{7\log(T)-2r_1}$$

From Line 7, $r_1 \leq 3\log(T)$. Thus:

$$N_i^{\lceil 7\log(T)\rceil} \geq N_i^1 T \geq T$$

If the $7\log(T)$'s round is reached and finished, then the horizon has arrived. Else, the algorithm will switch. ∎

**Lemma C.5** *Let $i$ be some arm, and $S$ be a series of steps. Denote $p_{min} = \min_{s \in S} p_i(s)$. Then, if $|S| \geq \frac{\log\left(\frac{1}{\delta}\right)}{p_{min}}$, w.p $1 - \delta$:*

$$\max_k \bar{L}_{S:k} - L_{S:k} \leq 2\sqrt{\frac{|S|\log\left(\frac{1}{\delta}\right)}{p_{min}}}$$

$$\max_k L_{S:k} - \bar{L}_{S:k} \leq 2\sqrt{\frac{|S|\log\left(\frac{1}{\delta}\right)}{p_{min}}}$$

**Proof:**

$$\mathbb{E}\left[\left(\bar{l}_i(t)\right)^2\right] = l_i(t)^2 \frac{p_i(t)}{p_i(t)^2} \leq \frac{1}{p_i(t)}$$

Thus, for every $k$ the variance is bounded by:

$$V\left(\bar{L}_{S:k} - L_{S:k}\right) \leq \frac{|S:k|}{p_{min}} \leq \frac{|S|}{p_{min}}$$

Using Lemma C.1 with $\sigma^2 = \frac{|S|}{p_{min}}$ and $R = \frac{1}{p_{min}}$:

$$\log\left(\Pr\left[\max_k \bar{L}_{S:k} - L_{S:k} \geq 2\sqrt{\frac{|S|\log\left(\frac{1}{\delta}\right)}{p_{min}}}\right]\right) \leq -\frac{4\frac{|S|\log\left(\frac{1}{\delta}\right)}{p_{min}}}{2\frac{|S|}{p_{min}} + \frac{4}{3}\frac{1}{p_{min}}\sqrt{\frac{|S|\log\left(\frac{1}{\delta}\right)}{p_{min}}}}$$

Since $|S| \geq \frac{\log\left(\frac{1}{\delta}\right)}{p_{min}}$:

$$\log\left(\Pr\left[\max_k \bar{L}_{S:k} - L_{S:k} \geq 2\sqrt{\frac{|S|\log\left(\frac{1}{\delta}\right)}{p_{min}}}\right]\right) \leq -\frac{4\frac{|S|\log\left(\frac{1}{\delta}\right)}{p_{min}}}{2\frac{|S|}{p_{min}} + \frac{4}{3}\frac{|S|}{p_{min}}} \leq -\log\left(\frac{1}{\delta}\right)$$

Which means that w.p $1 - \delta$:

$$\max_k \bar{L}_{S:k} - L_{S:k} \leq 2\sqrt{\frac{|S|\log\left(\frac{1}{\delta}\right)}{p_{min}}}$$

Which is exactly the first inequality in the lemma. The second has the same proof. ∎

# D    Stochastic

## D.1    Good Event

**Definition D.1** *Let $G_{sto}$ be the event that:*

$$\forall i \in [K], n \leq \left|\tilde{S}_i\right| \quad |\mu_i - \hat{\mu}_i(S_{:n})| \leq width_i(S_{:n})$$

$$\forall i \in [K], n \leq \left|\tilde{S}_i\right| \quad |\mu_i - \overline{\mu}_i(S_{:n})| \leq \overline{width}(S_{:n})$$

$$\forall i \in [K], n \leq \left|\tilde{S}_i\right|, S \in \left(\left\{\left(\tilde{S}_i\right)_{:n}\right\} \cup \left\{S_i^r | r \in [r_i(\bar{T})]\right\}\right) \quad \frac{1}{2}\sum_{s \in S} p_i(s) - 3\log(T) \leq n_i(S) \leq 2\sum_{s \in S} p_i(s) + 12\log(T)$$

$$\forall i \in [K] \quad E_i(\bar{T}) \leq \mathbb{E}\left[E_i(\bar{T})\right] + \sqrt{7}\log(T)$$

$$\forall S \quad \sum_{s \in S}[l_{a_s}(s) - l_{a^*}(s)] - \mathbb{E}\left[\sum_{s \in S}[l_{a_s}(s) - l_{a^*}(s)]\right] \leq 3\sqrt{T\log(T)}$$

*If the delays are stochastic, also:*

$$\sigma_{max} \leq 2\mathbb{E}[d] + 8\log(T)$$

**Lemma D.2** *For every state of $S$ and arm $i$, w.p $1 - \frac{2}{T}$:*

$$|\mu_i - \hat{\mu}_i(S)| \leq width_i(S)$$

**Proof:** Directly from Equation 1.6 of Slivkins [28]. ∎

**Lemma D.3** *With probability $1 - \frac{2}{T}$, For every arm $i$ and $n \leq \left|\tilde{S}_i\right|$:*

$$|\overline{\mu}_i(S_{:n}) - \mu_i| \leq \overline{width}(S_{:n})$$

**Proof:** Denote $S = S_{:n}$ for brevity.

If $|S| \leq 3K\log(T)$ then $\overline{width}(S) \geq 1$ and the bound is trivial.

Notice that since $n \leq \left|\tilde{S}_i\right|$, for $s \in S$ we have $p_i(s) \geq \frac{1}{K}$. If $|S| \geq 3K\log(T)$, w.p $1 - \frac{2}{T^3}$ from Lemma C.5:

$$|S||\overline{\mu}_i(S) - \mu_i| \leq 2\sqrt{3K|S|\log(T)}$$

$$|\overline{\mu}_i(t) - \mu_i| \leq 2\sqrt{\frac{3K\log(T)}{|S|}}$$

Union bound on all arms and $n \leq \left|\tilde{S}_i\right|$ we get the desired results. ∎

**Lemma D.4** *With probability* $1 - \frac{2}{T}$

$$\forall i \in [K], S \in \left( \left\{ \left( \tilde{S}_i \right)_{:k} \right\} \right) \vee S \in \{ S_i^r | r \in r_i(\bar{T}) \} \quad \frac{1}{2} \sum_{s \in S} p_i(s) - 3 \log(T) \le n_i(S) \le 2 \sum_{s \in S} p_i(s) + 12 \log(T)$$

**Proof:** Notice that $n_i(S)$ is a sum of bernoulli variables. Fix $S$ and $i$, from Lemmas C.2 and C.3, w.p $1 - \frac{2}{T^3}$:

$$\frac{1}{2} \sum_{s \in S} p_i(s) - 3 \log(T) \le n_i(t) \le 2 \sum_{s \in S} p_i(s) + 12 \log(T)$$

Union bound for all the options for $S$ and $i$ gives us the desired results. ∎

**Lemma D.5** *For every arm, w.p* $1 - \frac{1}{T}$:

$$E_i(\bar{T}) \le \mathbb{E}\left[E_i(\bar{T})\right] + \sqrt{7} \log(T)$$

**Proof:** Using Hoeffding, with probability $1 - \frac{1}{T^2}$:

$$E_i(\bar{T}) \le \mathbb{E}\left[E_i(\bar{T})\right] + \sqrt{\frac{1}{2} r_i(\bar{T}) \log(T^2)}$$

From Lemma C.4:

$$E_i(\bar{T}) \le \mathbb{E}\left[E_i(\bar{T})\right] + \sqrt{7} \log(T)$$

Union bound for all arms concludes the proof. ∎

**Lemma D.6** *For every state of S, w.p* $1 - \frac{1}{T}$:

$$\sum_{s \in S} [l_{a_s}(s) - l_{a^*}(s)] - \mathbb{E}\left[\sum_{s \in S} [l_{a_s}(s) - l_{a^*}(s)]\right] \le 3\sqrt{T \log(T)}$$

**Proof:** From Lemma C.1, with $\sigma^2 = T$ and $R = 1$:

$$\Pr\left[\exists S : \sum_{s \in S} [l_{a_s}(s) - l_{a^*}(s)] - \mathbb{E}\left[\sum_{s \in S} [l_{a_s}(s) - l_{a^*}(s)]\right] \ge 3\sqrt{T \log(T)}\right] \le exp\left(- \frac{9T \log(T)}{2T + 2\sqrt{T \log(T)}}\right)$$

$$\le exp(-\log(T)) = \frac{1}{T}$$

∎

**Lemma D.7** *If the delays are stochastic, with probability* $1 - \frac{1}{T}$:

$$\sigma_{max} \le 2\mathbb{E}[d] + 8 \log(T)$$

**Proof:** For every $t \le T$:

$$\mathbb{E}[\sigma(t)] = \sum_{s=1}^{t} \Pr[d > t - s]$$

$$= \sum_{s=1}^{t} \sum_{i=t-s+1}^{\infty} \Pr[d = i]$$

$$\le \sum_{i=1}^{\infty} i \Pr[d = i]$$

$$= \mathbb{E}[d]$$

Fix some $t \le T$. From Lemma C.3, w.p $1 - \frac{1}{T^2}$:

$$\sigma(t) \le 2\mathbb{E}[d] + 8 \log(T)$$

Union bound for all $t$ concludes the proof. ∎

**Corollary D.8** $G_{sto}$ *happens w.p* $1 - \frac{9}{T}$

**Proof:** Union bound of Lemmas D.2 to D.7 ∎

## D.2 Regret Analysis

**Lemma D.9** *Assume $G_{sto}$, the optimal arm $i^*$ will not be evicted.*

**Proof:** Assume by contradiction that arm $i^*$ was evicted. Namely, there is $S$ such that:
$$\text{ucb}^*(S) < \hat{\mu}_{i^*}(S) - 9\text{width}_{i^*}(S)$$

From $G_{sto}$:
$$\text{ucb}^*(S) < \mu_{i^*}(S) - 9\text{width}_{i^*}(S) < \mu^*$$

From the definition of ucb$^*$ and $G_{sto}$, there is an arm $i$ such that:
$$\mu_i \leq \text{ucb}^*(S) < \mu^*$$

Which contradicts the fact that $i^*$ is optimal. ∎

**Lemma D.10** *Assume $G_{sto}$, For every two arms $i_1, i_2$ and series $n \leq \min\left\{\left|\tilde{S}_{i_1}\right|, \left|\tilde{S}_{i_2}\right|\right\}$:*
$$width_{i_1}(S_{:n}) \leq 10width_{i_2}(S_{:n})$$

*Additionally, for every arm $i$:*
$$\tilde{\Delta}_i \leq \Delta_i \leq 2\tilde{\Delta}_i$$

**Proof:** Denote $S = S_{:n}$ for brevity.

Since $i_1$ was not eliminated:
$$\hat{\mu}_{i_1}(S) - 9\text{width}_{i_1}(S) \leq \text{ucb}^*(S) \leq \text{ucb}^*(S) \leq \text{ucb}_{i_2}(S) \leq \hat{\mu}_{i_2}(S) + \text{width}_{i_2}(S)$$

From $G_{sto}$:
$$\mu_i - 10\text{width}_{i_1}(S) \leq \mu^* + 2\text{width}_{i_2}(S)$$
$$\Delta_{i_1} \leq 10\text{width}_{i_1}(S) + 2\text{width}_{i_2}(S)$$

Notice that since $i_1$ and $i_2$ were not evicted in the steps of $S$, we have $\mathbb{E}[n_{i_1}(S)] = \mathbb{E}[n_{i_2}(S)]$. Thus:
$$n_{i_1}(S) \leq 2\mathbb{E}[n_{i_1}(S)] + 12\log(T)$$
$$\mathbb{E}[n_{i_1}(S)] \geq \frac{1}{2}n_{i_1}(S) - 6\log(T)$$
$$n_{i_2}(S) \geq \frac{1}{2}\mathbb{E}[n_{i_2}(S)] - 3\log(T)$$
$$\mathbb{E}[n_{i_2}(S)] \leq 2n_{i_2}(S) + 6\log(T)$$
$$\frac{1}{2}n_{i_1}(S) - 6\log(T) \leq 2n_{i_2}(S) + 6\log(T)$$
$$\frac{1}{4}n_{i_1}(S) - 24\log(T) \leq n_{i_2}(S)$$

If $n_i(S) \geq 192\log(T)$:
$$\frac{1}{4}n_{i_1}(S) - \frac{1}{8}n_{i_1}(S) \leq n_{i_2}(S)$$
$$\frac{1}{8}n_{i_1}(S) \leq n_{i_2}(S)$$
$$\sqrt{\frac{2\log(T)}{n_{i_2}(S)}} \leq 3\sqrt{\frac{2\log(T)(S)}{n_{i_1}(S)}}$$
$$\text{width}_{i_2}(S) \leq 3\text{width}_{i_1}(S)$$
$$\Delta_{i_1} \leq 16\text{width}_{i_1}(S)$$

If $n_i(S) \leq 192 \log(T)$:

$$\text{width}_{i_1}(S) \geq \sqrt{\frac{2 \log(T)}{192 \log(T)}} \geq \frac{1}{10}$$

$$\Delta_{i_1} \leq 1 \leq 10\text{width}_{i_1}(S)$$
$$\text{width}_{i_2}(S) \leq 1 \leq 10\text{width}_{i_1}(S)$$

In both cases:

$$\Delta_{i_1} \leq 16\text{width}_{i_1}(S) \leq 16\text{width}_{i_1}\left(\tilde{S}_{i_1}\right) = 2\tilde{\Delta}_{i_1}$$

Additionally, from $G_{sto}$:

$$\mu_{i_1} - 8\text{width}_{i_1}\left(\tilde{S}_{i_1}\right) \geq \hat{\mu}_{i_1}(\tilde{S}_{i_1}) - 9\text{width}_{i_1}\left(\tilde{S}_{i_1}\right) > \text{ucb}^*\left(\tilde{S}_{i_1}\right) \geq \mu^*$$
$$\Delta_{i_1} \geq 8\text{width}_{i_1}\left(\tilde{S}_{i_1}\right) = \tilde{\Delta}_{i_1}$$

$\blacksquare$

**Lemma D.11** *Let $N$ be a set over $\mathbb{R}$ with average $\mu$. Then, at most half of $N$ are greater than $2\mu$.*

**Proof:** Let $X$ be a r.v sampled from $N$ with uniform distribution. Easy to see that $\mathbb{E}[X] = \mu$. From Markov inequality:

$$Pr[X \geq 2\mu] \leq \frac{\mu}{2\mu} = \frac{1}{2}$$

$\blacksquare$

**Lemma D.12 (restatement of Lemma 4.2)** *If arm $i_1$ was eliminated before $i_2$ then,*

$$\Delta_{i_2} \leq 20\Delta_{i_1}$$

**Proof:** Let $i_1$ and $i_2$ be two arms such that $i_1$ was evicted before $i_2$. From Lemma D.10:

$$\Delta_{i_2} \leq 16\text{width}_{i_2}\left(\tilde{S}_{i_2}\right) \leq 16\text{width}_{i_2}\left(\tilde{S}_{i_1}\right) \leq 160\text{width}_{i_1}\left(\tilde{S}_{i_1}\right) \leq 20\Delta_{i_1}$$

$\blacksquare$

**Lemma D.13** *Assume $G_{sto}$, we have:*

$$\sum_i p_i^{max}(\tau_i)\Delta_i \leq \Delta_{max} \log(K)$$
$$\sum_i p_i^{max}(\tau_i)\Delta_i \leq 42\Delta_{avg} \log(K)$$

**Proof:**

Let $\mathcal{A}'$ be the state of $\mathcal{A}$ such that $|\mathcal{A}| = \frac{K}{2}$ (namely, the last half active arms). We show that:

$$\forall i \in \mathcal{A}' \quad \Delta_i \leq 40\Delta_{avg}$$

If $\Delta_i \leq 2\Delta_{avg}$ it is trivial. Otherwise, from Lemma D.11 there is an arm $j \notin \mathcal{A}'$ such that $\Delta_i \leq 2\Delta_{avg}$. From Lemma D.12, $\Delta_i \leq 20\Delta_j \leq 40\Delta_{avg}$.

Notice that if arm $i$ was the $j$th evicted arm, we have:

$$p_i^{max}(\tau_i) \leq \frac{1}{j}$$

Which means:

$$\sum_i p_i^{max}(\tau_i)\Delta_i = \sum_{i\notin\mathcal{A}'} p_i^{max}(\tau_i)\Delta_i + \sum_{i\in\mathcal{A}'} p_i^{max}(\tau_i)\Delta_i$$

$$\leq \sum_{i\notin\mathcal{A}'} \frac{\Delta_i}{\frac{K}{2}} + \sum_{j=1}^{\frac{K}{2}} \frac{40\Delta_{avg}}{j}$$

$$\leq 2\Delta_{avg} + 40\Delta_{avg}\log(K)$$

∎

**Lemma D.14** *Assume $G_{sto}$, then for any arm $i$ and step $t < \tau_i$:*

$$m_i(t)\Delta_i \leq \frac{515\log(T)}{\Delta_i} + 2\sigma(t)\Delta_i p_i^{max}(\tau_i)$$

**Proof:** Let $S$ be the set of observed pulls at time $t$. From Lemma D.10:

$$\Delta_i \leq 16\sqrt{\frac{2\log(T)}{n_i(S)}}$$

$$n_i(S) \leq \frac{512\log(T)}{\tilde{\Delta}^2}$$

At step $t$, there are $\sigma(t)$ missing pulls. When those missing pulls were pull, the probability of arm $i$ was bounded by $p_i^{max}(\tau_i - 1)$ (as it is its general maximum probability). Thus, from $G_{sto}$, we have:

$$m_i(t) \leq n_i(S) + 2\sigma(t)p_i^{max}(\tau_i - 1) + 3\log(T)$$

Thus:

$$m_i(t)\Delta_i \leq \frac{515\log(T)}{\Delta_i} + 2\sigma(t)p_i^{max}(\tau_i)\Delta_i$$

∎

**Lemma D.15** *Assume $G_{sto}$ and that $K \leq \frac{T}{12\log(T)}$, then for any arm $i$ and step $\tau_i < t \leq \bar{T}$:*

$$(m_i(t) - m_i(\tau_i))\Delta_i \leq \frac{71728\log(T)}{\Delta_i} + 8p_i^{max}(\tau_i)\sigma_{max}(t)\Delta_i$$

**Proof:** From the algorithm defintion we have:

$$m_i(t) - m_i(\tau_i) = \sum_{r=1}^{r_i(t)} n_i(S_i^r) + \sum_{j=0}^{\log(T)} n_i(M_i^{p_i^1 2^{-j}}(t) \setminus C_i^{p_i^1 2^{-j}}(t)) \tag{6}$$

From $G_{sto}$, for every $i$ and every phase $r_i$:

$$\mathbb{E}[n_i(S_i^r)] = p_i^r N_i^r = p_i^1 N_i^1 = \frac{1280}{\tilde{\Delta}_i^2}$$

From Lemma C.4 and $G_{sto}$:

$$\mathbb{E}\left[\sum_{r=1}^{r_i(t)} n_i(S_i^r)\right] \leq 7\log(T)\frac{1280}{\tilde{\Delta}_i^2} \leq \frac{8960\log(T)}{\tilde{\Delta}_i^2}$$

$$\sum_{r=1}^{r_i(t)} n_i(S_i^r) \leq \frac{17920\log(T)}{\tilde{\Delta}_i^2} + 3\log(T) \leq \frac{17923\log(T)}{\tilde{\Delta}_i^2}$$

Let $s \leq t$ be the last pull of a phase w.p $p$, which means that $B_i^p(s) \setminus C_i^p(s) = \emptyset$, so $|M_i^p(t) \setminus C_i^p(t)| = |M_i^p(s) \setminus C_i^p(s)| \leq \sigma(s) \leq \sigma_{max}(t)$. Then:

$$\mathbb{E}[n_i(M_i^p \setminus C_i^p)] \leq p\sigma_{max}(t)$$

$$\sum_{j=0}^{\log(T)} \mathbb{E}\left[n_i(M_i^{p_i^1 \cdot 2^{-j}} \setminus C_i^{p_i^1 2^{-j}})\right] \leq \sum_{j=0}^{\log(T)} p_i^1 2^{-j}\sigma_{max}(t) \leq 2p_i^1\sigma_{max}(t)$$

$$\sum_{j=0}^{\log(T)} n_i(M_i^{p_i^1 2^{-j}} \setminus C_i^{p_i^1 2^{-j}}) \leq 4p_i^1\sigma_{max}(t) + 3\log(T) \qquad (G_{sto})$$

Thus, from Equation (6):

$$m_i(t) - m_i(\tau_i) \leq \frac{17926\log(T)}{\tilde{\Delta}_i^2} + 4p_i^1\sigma_{max}(t)$$

From $G_{sto}$ and the assumption that $K \leq \frac{T}{12\log(T)}$:

$$\frac{1}{K} \leq p_i^{max}(\tau_i)$$

$$n_i(\tilde{S}_i) \leq 2\left|\tilde{S}_i\right|p_i^{max}(\tau_i) + 12\log(T) \leq 2Tp_i^{max}(\tau_i) + \frac{T}{K} \leq 3Tp_i^{max}(\tau_i)$$

$$p_i^1 = \frac{1}{2K} + \frac{n_i(\tilde{S}_i)}{2T} \leq \frac{p_i^{max}(\tau_i)}{2} + \frac{3p_i^{max}(\tau_i)}{2} = 2p_i^{max}(\tau_i)$$

$$m_i(t) - m_i(\tau_i) \leq \frac{17932\log(T)}{\tilde{\Delta}_i^2} + 8p_i^{max}(\tau_i)\sigma_{max}(t)$$

From Lemma D.10:

$$m_i(t) - m_i(\tau_i) \leq \frac{71728\log(T)}{\Delta_i^2} + 8p_i^{max}(\tau_i)\sigma_{max}(t)$$

$$(m_i(t) - m_i(\tau_i))\Delta_i \leq \frac{71728\log(T)}{\Delta_i} + 8p_i^{max}(\tau_i)\sigma_{max}(t)\Delta_i$$

∎

**Corollary D.16** *Assume $G_{sto}$, for every $t \leq \bar{T}$:*

$$\sum_{i=1}^{K} \Delta_i m_i(t) \leq \sum_{i=1}^{K} \frac{72243\log(T)}{\Delta_i} + 10\sigma_{max}(t)\Delta_{max}\log(K)$$

$$\sum_{i=1}^{K} \Delta_i m_i(t) \leq \sum_{i=1}^{K} \frac{72243\log(T)}{\Delta_i} + 420\sigma_{max}(t)\Delta_{avg}\log(K)$$

**Proof:** If $K \leq \frac{T}{12\log(T)}$, it follows directly from Lemmas D.13 to D.15. Else, we have:

$$\sum_{i=1}^{K} \Delta_i m_i(t) \leq T \leq 12K\log(T) \leq \sum_{i=1}^{K} \frac{12\log(T)}{\Delta_i}$$

∎

## D.3  With high probability the algorithm doesn't switch

**Lemma D.17** *Assume $G_{sto}$, for every arm $i$ and $S \subseteq \tilde{S}_i$:*
$$\overline{\mu}_i(S) \in [\overline{lcb}_i(S) - \overline{width}(S), \overline{ucb}_i(S) + \overline{width}(S)]$$

**Proof:** From $G_{sto}$:

$$\overline{\text{lcb}}_i(S) \leq \mu_i \leq \overline{\mu}_i(S) + \overline{\text{width}}(S)$$
$$\overline{\text{ucb}}_i(S) \geq \mu_i \geq \overline{\mu}_i(S) - \overline{\text{width}}(S)$$

∎

**Lemma D.18** *Assume $G_{sto}$, for every state of $S$ at time $t$:*

$$\sum_{s \in S} [l_{a_s}(s) - \text{ucb}^*(S)] \leq 272\sqrt{KT \log(T)} + 30\sigma_{max}(t)\log(K)$$

**Proof:** From Corollary D.16:

$$\mathbb{E}\left[\sum_{s \in S} [l_{a_s}(s) - l_{a^*}(s)]\right] \leq \sum_{i=1}^{K} \frac{72243\log(T)}{\Delta_i} + 30\sigma_{max}(t)\Delta_i$$

$$\leq \sum_{i \text{ s.t } \Delta_i \leq 269\sqrt{\frac{\log(T)}{KT}}} 269\sqrt{\frac{T\log(T)}{K}} + \sum_{i \text{ s.t } \Delta_i \geq 269\sqrt{\frac{\log(T)}{KT}}} \frac{72243\log(T)}{\Delta_i} + 30\sigma_{max}(t)$$

$$\leq 269\sqrt{KT\log(T)} + 30\sigma_{max}(t)\log(K)$$

From $G_{sto}$, for every state of $S$:

$$\sum_{s \in S} [l_{a_s}(s) - \text{ucb}^*(S)] \leq \sum_{s \in S} [l_{a_s}(s) - l_{a^*}(s)] \leq 272\sqrt{KT\log(T)} + 30\sigma_{max}(t)\log(K)$$

∎

**Lemma D.19** *Assume $G_{sto}$, for every arm $i$, $E_i(\bar{T}) \leq 3\log(T)$.*

**Proof:** Fix phase $r$. We will show that w.p $\frac{31}{32}$,

$$|S_i^r|\tilde{\mu}_i - \bar{L}_i^r \leq \frac{1}{4}\tilde{\Delta}N_i^r$$

If $|S_i^r| \leq \frac{1}{4}\tilde{\Delta}_i N_i^r$, this is trivial. Otherwise, it means that:

$$|S_i^r| \geq \frac{1}{4}\tilde{\Delta}_i N_i^r$$
$$= \frac{1}{4}\tilde{\Delta}_i \frac{p_i^1 N_i^1}{p_i^r}$$
$$= \frac{1}{4}\tilde{\Delta}_i \frac{1280}{p_i^r \tilde{\Delta}_i^2}$$
$$= \frac{320}{p_i^r \tilde{\Delta}_i}$$
$$\geq \frac{40}{p_i^r} \qquad\qquad (\tilde{\Delta}_i \leq 8)$$

Now we can use Lemma C.5, w.p $\frac{31}{32}$, as the inequality $|S_i^r| \geq \frac{\log(32)}{p_i^r}$ is satisfied. We have:

$$\mathbb{E}[L_i(S_i^r)] - \bar{L}_i(S_i^r) \leq 2\sqrt{\frac{5N_i^r}{p_i^r}}$$
$$= 2\sqrt{\frac{5N_i^{r\,2}\tilde{\Delta}_i^2}{1280}}$$
$$= \frac{1}{8}N_i^r\tilde{\Delta}_i \qquad\qquad (7)$$

Additionally, from $G_{sto}$:

$$\tilde{\mu}_i - \mu_i \leq \text{width}_i\left(\tilde{S}_i\right) = \frac{1}{8}\tilde{\Delta} \qquad\qquad (G_{sto})$$

$$|S_i^r|\tilde{\mu}_i - \mathbb{E}[L_i^r] \leq \frac{1}{8}\tilde{\Delta}|S_i^r| \leq \frac{1}{8}\tilde{\Delta}N_i^r$$

$$|S_i^r|\tilde{\mu}_i - \bar{L}_i^r \leq \frac{1}{4}\tilde{\Delta}N_i^r \qquad\qquad (\text{Equation (7)})$$

All of the above is true to all states throughout the phase (since Lemma C.5 is true for $\max_k$).

This means that in every phase Line 6 happens w.p $\frac{1}{32}$. Thus:

$$\mathbb{E}\big[E_i(\bar{T})\big] = \frac{r_i(\bar{T})}{32}$$

$$\leq \frac{7\log(T)}{32} \qquad\qquad (\text{Lemma C.4})$$

From $G_{sto}$:

$$E_i(\bar{T}) \leq \mathbb{E}\big[E_i(\bar{T})\big] + \sqrt{7}\log(T) \leq 3\log(T)$$

■

**Corollary D.20** *Assume $G_{sto}$, $T = \bar{T}$.*

**Proof:** Directly from Lemmas D.17 to D.19

■

### D.4 Conclusion

**Theorem D.21** *For adversarial delays:*

$$\mathcal{R}_{sto} \leq O\left(\sum_{i=1}^{K}\frac{\log(T)}{\Delta_i} + \sigma_{max}\Delta_{avg}\log(K)\right)$$

*For stochastic delays we can also say:*

$$\mathcal{R}_{sto} \leq O\left(\sum_{i=1}^{K}\frac{\log(T)}{\Delta_i} + \mathbb{E}[d]\Delta_{avg}\log(K)\right)$$

**Proof:** Assume $G_{sto}$, from Corollaries D.16 and D.20, The above is true with probability $1 - \frac{2}{T}$.

From Corollary D.8, this is asymptotically true even without the assumption of $G_{sto}$. ■

## E  Adversarial

**Lemma E.1** *Let $X$ be a random variable such that for every $x \geq 0$ there is some $a > 0$ such that:*

$$F_X(x) \geq 1 - e^{-x/a}$$

*Then:*

$$\mathbb{E}[X] \leq a$$

**Proof:** We use the CDF representation of the expectation:

$$\mathbb{E}[X] = \int_0^{\infty}(1 - F(x))dx + \int_{-\infty}^{0}F(x)dx$$

$$\leq \int_0^{\infty}(1 - F(x))dx$$

$$\leq \int_0^{\infty}e^{-x/a}dx$$

$$= \left[-ae^{-x/a}\right]_0^{\infty}$$

$$= a$$

■

**Lemma E.2 (restatement of Lemma 5.2)** *Let $\mathcal{H}_i^r$ be the history (i.e., chosen actions) of rounds that are observed by the begining of the $r$ phase of arm $i$. Denote $\mathbb{E}_i^r[\cdot] = \mathbb{E}[\cdot \mid \mathcal{H}_i^r]$ and $\Pr_i^r[\cdot] = \Pr[\cdot \mid \mathcal{H}_i^r]$. For every arm $i$ and phase $r$ we have:*

$$\mathbb{E}_i^r\left[|S_i^r|\mathrm{ucb}^*\left(\tilde{S}_i\right) - L_i(S_i^r)\right] \leq \frac{3}{8}\tilde{\Delta}_i N_i^r - \frac{9}{8}\tilde{\Delta}_i \mathbb{E}_i^r[|S_i^r|]$$

**Proof:** Let $S$ be the sequence of $N_i^r$ observed rounds starting from the beginning of $r$ phase of arm $i$, so that the first $|S_i^r|$ rounds in $S$ are exactly $S_i^r$. Let also $X_1, ..., X_{N_i^r} \stackrel{i.i.d}{\sim}$ Bernoulli$(p_i^r)$. From Lemma C.5, if $N_i^r \geq \frac{m}{p_i^r}$, we have for every $m > 0$, conditioned on the history $\mathcal{H}_i^r$, with probability of at least $1 - e^{-m}$:

$$\max_k \sum_{t \in S_{:k}} \left[\frac{X_t l_t}{p_i^r} - l_i(t)\right] \leq 2\sqrt{\frac{mN_i^r}{p_i^r}}$$

$$\leq 2\sqrt{\frac{mN_i^r}{\frac{p_i^1 N_i^1}{N_i^r}}}$$

$$= 2\sqrt{\frac{mN_i^{r\,2}\tilde{\Delta}_i^2}{1280}}$$

$$\leq \frac{mN_i^r\tilde{\Delta}_i}{\sqrt{320}}$$

Now, note that for $k = |S_i^r|$, $\sum_{t \in S_{:k}}\left[\frac{X_t l_t}{p_i^r} - l_i(t)\right]$ distributes exactly like $L_i(S_i^r) - \bar{L}_i(S_i^r)$. Thus, conditioned on the history $\mathcal{H}_i^r$, with probability of at least $1 - e^{-m}$:

$$\bar{L}_i(S_i^r) - L_i(S_i^r) \leq |S_i^r| \leq \frac{mN_i^r\tilde{\Delta}_i}{\sqrt{320}}$$

If $N_i^r \leq \frac{m}{p_i^r}$:

$$\bar{L}_i(S_i^r) - L_i(S_i^r) \leq N_i^r$$

$$\leq \frac{m}{p_i^r}$$

$$= \frac{mN_i^r}{p_i^1 N_i^1}$$

$$= \frac{mN_i^r\tilde{\Delta}_i^2}{1280}$$

$$\leq \frac{mN_i^r\tilde{\Delta}_i}{160} \qquad (\tilde{\Delta}_i \leq 8)$$

$$\leq \frac{mN_i^r\tilde{\Delta}_i}{\sqrt{320}}$$

From Lemma E.1:

$$\mathbb{E}_i^r\left[\bar{L}_i(S_i^r) - L_i(S_i^r)\right] \leq \frac{N_i^r\tilde{\Delta}_i}{\sqrt{320}} \leq \frac{1}{8}N_i^r\tilde{\Delta}_i \qquad (8)$$

Notice that if the phase was not finished it means that $|S_i^r|\tilde{\mu}_i - \bar{L}_i^r \leq \frac{1}{4}\tilde{\Delta}N_i^r$. Given that we have:

$$\mathbb{E}_i^r\left[|S_i^r|\mathrm{ucb}^*\left(\tilde{S}_i\right) - \bar{L}_i(S_i^r) + \frac{9}{8}\tilde{\Delta}_i|S_i^r|\right] = \mathbb{E}_i^r\left[|S_i^r|\left(\mathrm{ucb}^*\left(\tilde{S}_i\right) + 9\mathrm{width}_i\left(\tilde{S}_i\right)\right) - \bar{L}_i(S_i^r)\right]$$

$$< \mathbb{E}_i^r\left[|S_i^r|\tilde{\mu}_i - \bar{L}_i(S_i^r)\right] \qquad \text{(Elimination inequality)}$$

$$\leq \frac{1}{4}\tilde{\Delta}_i N_i^r$$

Combining the above with Equation (8) completes the proof.

∎

**Lemma E.3** *For every arm $i$ and phase $r$,*

$$\mathbb{E}_i^r\left[\sum_{r'=r}^{r_i(T)}\left[\left|S_i^{r'}\right|\mathrm{ucb}^*\left(\tilde{S}_i\right) - L_i(S_i^{r'})\right]\right] \leq \frac{3(r_i(T) - r - 2)}{8}\tilde{\Delta}_i N_i^1 + \frac{3}{4}\tilde{\Delta}_i N_i^r$$

**Proof:** We will prove using reverse induction on $r$.

For $r = r_i(T)$ (namely, after all phases) we have:

$$\mathbb{E}_i^r\left[\sum_{r'=r_i(T)}^{r_i(T)}\left[\left|S_i^{r'}\right|\mathrm{ucb}^*\left(\tilde{S}_i\right) - L_i^{r'}\right]\right] = 0$$

$$\frac{3(0 - 2)}{8}\tilde{\Delta}_i N_i^1 + \frac{3}{4}\tilde{\Delta}_i N_i^r \geq \frac{3(0 - 2)}{8}\tilde{\Delta}_i N_i^1 + \frac{3}{4}\tilde{\Delta}_i N_i^1 = 0$$

Assume true for $r + 1$, we prove for $r$.

If Line 11 was triggered it means that $|S_i^r| = N_i^r$ and $N_i^{r+1} = 2N_i^r$. We have:

$$\mathbb{E}_i^r\left[|S_i^r|\mathrm{ucb}^*\left(\tilde{S}_i\right) - L_i^r\right] \leq \frac{3}{8}\tilde{\Delta}_i N_i^r - \frac{9}{8}\tilde{\Delta}_i N_i^r \qquad \text{(Lemma E.2)}$$

$$\mathbb{E}_i^r\left[\sum_{r'=r}^{r_i(T)}\left[\left|S_i^{r'}\right|\mathrm{ucb}^*\left(\tilde{S}_i\right) - L_i(S_i^{r'})\right]\right] = \mathbb{E}_i^r\left[\sum_{r'=r+1}^{r_i(T)}\left[\left|S_i^{r'}\right|\mathrm{ucb}^*\left(\tilde{S}_i\right) - L_i(S_i^{r'})\right]\right] + \mathbb{E}_i^r\left[|S_i^r|\mathrm{ucb}^*\left(\tilde{S}_i\right) - L_i^r\right]$$

$$\leq \frac{3(r_i(T) - r - 3)}{8}\tilde{\Delta}_i N_i^1 + \frac{3}{4}\tilde{\Delta}_i N_i^{r+1} - \frac{3}{4}\tilde{\Delta}_i N_i^r$$

$$\leq \frac{3(r_i(T) - r - 2)}{8}\tilde{\Delta}_i N_i^1 + \frac{3}{2}\tilde{\Delta}_i N_i^r - \frac{3}{4}\tilde{\Delta}_i N_i^r$$

$$= \frac{3(r_i(T) - r - 2)}{8}\tilde{\Delta}_i N_i^1 + \frac{3}{4}\tilde{\Delta}_i N_i^r$$

If Line 6 was triggered and $N_i^r \neq N_i^1$ it means that $N_i^{r+1} = \frac{1}{2}N_i^r$. We have:

$$\mathbb{E}_i^r\left[|S_i^r|\mathrm{ucb}^*\left(\tilde{S}_i\right) - L_i^r\right] \leq \frac{3}{8}\tilde{\Delta}_i N_i^r \qquad \text{(Lemma E.2)}$$

$$\mathbb{E}_i^r\left[\sum_{r'=r}^{r_i(T)}\left[\left|S_i^{r'}\right|\mathrm{ucb}^*\left(\tilde{S}_i\right) - L_i(S_i^{r'})\right]\right] = \mathbb{E}_i^r\left[\sum_{r'=r+1}^{r_i(T)}\left[\left|S_i^{r'}\right|\mathrm{ucb}^*\left(\tilde{S}_i\right) - L_i(S_i^{r'})\right]\right] + \mathbb{E}_i^r\left[|S_i^r|\mathrm{ucb}^*\left(\tilde{S}_i\right) - L_i^r\right]$$

$$\leq \frac{3(r_i(T) - r - 3)}{8}\tilde{\Delta}_i N_i^1 + \frac{3}{4}\tilde{\Delta}_i N_i^{r+1} + \frac{3}{8}\tilde{\Delta}_i N_i^r$$

$$\leq \frac{3(r_i(T) - r - 2)}{8}\tilde{\Delta}_i N_i^1 + \frac{3}{8}\tilde{\Delta}_i N_i^r + \frac{3}{8}\tilde{\Delta}_i N_i^r$$

$$= \frac{3(r_i(T) - r - 2)}{8}\tilde{\Delta}_i N_i^1 + \frac{3}{4}\tilde{\Delta}_i N_i^r$$

If $N_i^r = N_i^1$ it means that $N_i^{r+1} = N_i^r$. We have:

$$\mathbb{E}_i^r \left[ |S_i^r| \text{ucb}^* \left( \tilde{S}_i \right) - L_i^r \right] \le \frac{3}{8} \tilde{\Delta}_i N_i^1 \qquad \text{(Lemma E.2)}$$

$$\mathbb{E}_i^r \left[ \sum_{r'=r}^{r_i(T)} \left[ \left| S_i^{r'} \right| \text{ucb}^* \left( \tilde{S}_i \right) - L_i(S_i^{r'}) \right] \right] = \mathbb{E}_i^r \left[ \sum_{r'=r+1}^{r_i(T)} \left[ \left| S_i^{r'} \right| \text{ucb}^* \left( \tilde{S}_i \right) - L_i(S_i^{r'}) \right] \right] + \mathbb{E}_i^r \left[ |S_i^r| \text{ucb}^* \left( \tilde{S}_i \right) - L_i^r \right]$$

$$\le \frac{3(r_i(T) - r - 3)}{8} \tilde{\Delta}_i N_i^1 + \frac{3}{4} \tilde{\Delta}_i N_i^{r+1} + \frac{3}{8} \tilde{\Delta}_i N_i^1$$

$$= \frac{3(r_i(T) - r - 2)}{8} \tilde{\Delta}_i N_i^1 + \frac{3}{4} \tilde{\Delta}_i N_i^r$$

∎

**Lemma E.4** *For every arm $i$:*

$$\mathbb{E} \left[ \sum_{t=\tau_i}^{\bar{T}} \left[ \text{ucb}^* \left( \bar{S} \right) - l_i(t) \right] \right] \le 594 \sqrt{KT \log(T)}$$

**Proof:** Lemma E.3 with $r = 1$ gives:

$$\mathbb{E}_i^1 \left[ \sum_{r'=1}^{r_i(T)} \left[ \left| S_i^{r'} \right| \text{ucb}^* \left( \tilde{S}_i \right) - L_i(S_i^{r'}) \right] \right] \le \frac{3(r_i(T) - 3)}{8} \tilde{\Delta}_i N_i^1 + \frac{3}{4} \tilde{\Delta}_i N_i^1 \le \frac{3 r_i(T)}{8} \tilde{\Delta}_i N_i^1$$

From Lemma C.4:

$$\mathbb{E}_i^1 \left[ \sum_{r'=1}^{r_i(T)} \left[ \left| S_i^{r'} \right| \text{ucb}^* \left( \tilde{S}_i \right) - L_i(S_i^{r'}) \right] \right] \le \frac{21 \log(T)}{8} \tilde{\Delta}_i N_i^1 \qquad (9)$$

If $\frac{n_i(\tilde{S}_i)}{T} \ge \frac{1}{K}$:

$$p_i^1 \tilde{\Delta}_i \ge \frac{n_i(\tilde{S}_i)}{2T} 8 \sqrt{\frac{2 \log(T)}{n_i(\tilde{S}_i)}}$$

$$= \sqrt{\frac{32 \log(T)}{T}} \sqrt{\frac{n_i(\tilde{S}_i)}{T}}$$

$$\ge \sqrt{\frac{32 \log(T)}{KT}}$$

If $\frac{n_i(\tilde{S}_i)}{T} \le \frac{1}{K}$:

$$p_i^1 \tilde{\Delta}_i \ge \frac{8}{2K} \sqrt{\frac{2 \log(T)}{n_i(\tilde{S}_i)}}$$

$$= \sqrt{\frac{32 \log(T)}{K}} \sqrt{\frac{1}{K n_i(\tilde{S}_i)}}$$

$$\ge \sqrt{\frac{32 \log(T)}{KT}}$$

In any case:

$$p_i^1 \tilde{\Delta}_i \ge \sqrt{\frac{32 \log(T)}{KT}} \qquad (10)$$

Since for every $i$, $\tilde{S}_i$ is a sub-series of $\bar{S}$, we have $\mathrm{ucb}^*\left(\tilde{S}_i\right) \geq \mathrm{ucb}^*(\bar{S})$. From Equations (9) and (10):

$$\mathbb{E}_i^1\left[\sum_{t=\tau_i}^{\bar{T}}\left[\mathrm{ucb}^*(\bar{S}) - l_i(t)\right]\right] \leq \mathbb{E}_i^1\left[\sum_{t=\tau_i}^{\bar{T}}\left[\mathrm{ucb}^*\left(\tilde{S}_i\right) - l_i(t)\right]\right]$$

$$\leq \frac{21\log(T)}{8}\tilde{\Delta}_i N_i^1$$

$$= \frac{3360\log(T)}{p_i^1\tilde{\Delta}_i}$$

$$\leq 594\sqrt{KT\log(T)}$$

$\blacksquare$

**Theorem E.5**

$$R_{adv} \leq O\left(\sqrt{KT\log(T)} + \log(K)\sigma_{max} + R_{ALG}\right)$$

**Proof:** From Line 3:

$$\sum_{s\in\bar{S}}[l_{a_s}(s) - \mathrm{ucb}^*(\bar{S})] \leq 272\sqrt{KT\log(T)} + 10\sigma_{max}\log(K)$$

$$\sum_{t=1}^{\bar{T}}[l_{a_t}(t)] - \bar{T}\mathrm{ucb}^*(\bar{S}) \leq 272\sqrt{KT\log(T)} + 11\sigma_{max}\log(K) \qquad (11)$$

From Wald's equation and Line 1, for every arm $i$:

$$\mathbb{E}\left[\sum_{t\in\tilde{S}_i} l_i(t)\right] = \mathbb{E}\left[\left|\tilde{S}_i\right|\overline{\mu}_i\left(\tilde{S}_i\right)\right]$$

$$\geq \mathbb{E}\left[\left|\tilde{S}_i\right|\overline{\mathrm{lcb}}_i\left(\tilde{S}_i\right) - \overline{\mathrm{width}}\left(\tilde{S}_i\right)\right]$$

$$\geq \mathbb{E}\left[\left|\tilde{S}_i\right|\left(\overline{\mathrm{ucb}}_i\left(\tilde{S}_i\right) - 3\overline{\mathrm{width}}\left(\tilde{S}_i\right)\right)\right]$$

$$\geq \mathbb{E}\left[\left|\tilde{S}_i\right|\left(\mathrm{ucb}^*\left(\tilde{S}_i\right) - 3\sqrt{\frac{2K\log(T)}{\left|\tilde{S}_i\right|}}\right)\right]$$

$$\geq \mathbb{E}\left[\left|\tilde{S}_i\right|\mathrm{ucb}^*(\bar{S}) - 3\sqrt{2KT\log(T)}\right]$$

Adding the missing pulls we get:

$$\mathbb{E}\left[\left|\tilde{S}_i\right|\mathrm{ucb}^*(\bar{S}) - \sum_{t=1}^{\tau_i-1} l_i(t)\right] \leq 3\sqrt{2KT\log(T)} + \sigma(\tau_i - 1) \qquad (12)$$

From Equations (11) and (12) and lemma E.4, for every arm $i$:

$$\mathbb{E}\left[\sum_{t=1}^{\bar{T}}[l_{a_t}(t) - l_i(t)]\right] = \mathbb{E}\left[\sum_{t=1}^{\bar{T}}[l_{a_t}(t)] - \bar{T}\mathrm{ucb}^*(\bar{S})\right]$$

$$+ \mathbb{E}\left[(\tau_i - 1)\mathrm{ucb}^*(\bar{S}) - \sum_{t=1}^{\tau_i-1} l_i(t)\right]$$

$$+ \mathbb{E}\left[\sum_{t=\tau_i}^{\bar{T}}\left[\mathrm{ucb}^*(\bar{S}) - l_i(t)\right]\right]$$

$$\leq 869\sqrt{KT\log(T)} + 12\sigma_{max}\log(K)$$

Since after $\bar{T}$ the algorithm switches to ALG, we have:

$$\mathbb{E}\left[\sum_{t=\bar{T}+1}^{T}\left[l_{a_t}(t) - l_i(t)\right]\right] \leq R_{\text{ALG}}$$

Which concludes the proof. ∎

**Lemma E.6 (Restatement of Lemma 5.3)**

$$\sigma_{max} \leq O\left(\min_{S \in [T]}\left\{|S| + \sqrt{D_{\bar{S}}}\right\}\right)$$

**Proof:** Let $S^*$ be the set that minimizes $|S| + \sqrt{D_{\bar{S}}}$. If $|S^*| \geq \frac{1}{2}\sigma_{max}$ it concludes the proof. Continuing with the case that $|S^*| < \frac{1}{2}\sigma_{max}$.

Let $t$ be the step such that $\sigma(t) = \sigma_{max}$. Since $|S^*| < \frac{1}{2}\sigma_{max}$, after skipping there are at least $\frac{1}{2}\sigma_{max}$ non-skipped missing steps at time $t$. Let $s_1, ..., s_{\frac{1}{2}\sigma(t)} \in \bar{S}^*$ be the series of those $\frac{1}{2}\sigma(t)$ missing steps, ordered in descending order of when they were pulled. Namely, $s_1$ is the most recent pull in the series and $s_{\frac{1}{2}\sigma(t)}$ is the oldest pull.

Since there are at least $i-1$ missing pulls that were pulled after $s_i$, we have $t - s_i \geq i$. Additionally, since $s_i$ is missing, we have $s_i + d_{s_i} > t$. Combining both we have $d_{s_i} > i$. Thus:

$$D_{\bar{S}^*} \geq \sum_{i=1}^{\frac{1}{2}\sigma(t)} d_{s_i} > \sum_{i=1}^{\frac{1}{2}\sigma(t)} i \geq \frac{\left(\frac{1}{2}\sigma(t)\right)^2}{2} = \frac{1}{8}\sigma(t)^2$$

∎

# F  Removing the $\log(K)$ factor

We show that a simple modification of the algorithm can eliminate the $\log(K)$ factor from the additive delay term in both the adversarial and stochastic settings (Theorems D.21 and E.5). To avoid adding complexity to the already intricate algorithm, we present this modification separately as an optional, opt-in feature.

---

**Algorithm 8** Delayed SAPO Algorithm with reduced $\log(K)$

---

**Require:** Number of arms $K$, number of rounds $T \geq K$, Algorithm ALG.
 1: Initialize active arms $\mathcal{A} = \{1, \ldots, K\}$, $S = \langle\rangle$, $h = 1$, $G = \emptyset$
 2: **for** $t = 1, 2, \ldots, T$ **do**
 3:     **for** $s \in B \setminus S$ **do**
 4:         $S = S + \langle s \rangle$
 5:         **if** not BSC($S$) (Procedure 7) **then**
 6:             └ Switch to ALG.
 7:         $U(t) = \{i \in \mathcal{A} : \hat{\mu}_i(S) - 9\text{width}_i(S) > \text{ucb}^*(S)\}$  ▷ *Ghosting*
 8:         $G = G \cup U$
 9:         **for** $i \in U$ **do**  ▷ *Initialization for phases variables*
10:             Set $p_i^1 = \frac{1}{2K} + \frac{n_i(S)}{2T}$, $\tilde{S}_i = S$, $\tilde{\mu}_i = \hat{\mu}_i(S)$, $\tilde{\Delta}_i = 8\text{width}_i(S)$, $N_i^1 := 1280/(p_i^1\tilde{\Delta}_i^2)$,
                └ $E_i = 0$, $r_i = 1$, $S_i^1 = \langle\rangle$, $C_i^{p_i^1 \cdot 2^{-j}} = \emptyset \; \forall j \in [\log(T)]$
11:         **if** $\min_i \text{width}_i(S) \leq 2^{-h}$ **then**  ▷ *Elimination point*
12:             **for** $i \in G$ **do**
13:                 └ $\tau_i = t$, $S_i^g = S \setminus \tilde{S}_i$
14:             └ $\mathcal{A} = \mathcal{A} \setminus G$, $G = \emptyset$, $h = h + 1$
15:     **for** $i \in ([K] \setminus \mathcal{A})$ **do**
16:         $p_i(t), err = \text{EAP}(i)$ (Procedure 6)
17:         **if** $err$ **then**
18:             └ └ Switch to ALG.
19:     $\forall i \in \mathcal{A} \quad p_i(t) = \left(1 - \sum_{j \in ([K] \setminus \mathcal{A}(t))} p_j(t)\right)/|\mathcal{A}(t)|$
20:     └ Observe feedback and update variables

---

**Procedure 9** Basic Stochastic Checks (BSC) Subroutine with reduced $\log(K)$

**Require:** Series of processed pulls $S$
1: **if** $\exists i \in \mathcal{A} : \overline{\mu}_i(S) \notin [\overline{\text{lcb}}_i(S) - \overline{\text{width}}(S), \overline{\text{ucb}}_i(S) + \overline{\text{width}}(S)]$ **then**
2: $\quad \llcorner$ **return** False
3: **if** $\sum_{s' \in S} (l_{a_{s'}}(s') - \text{ucb}^*(S)) > C\left(\sqrt{KT \log(T)} + \sigma_{max}(t)\right)$ **then**
4: $\quad \llcorner$ **return** False
  $\quad$ **return** True

The key change involves introducing elimination points, where the $h$th elimination point is when the confidence *width* of at least one arm falls below $2^{-h}$. When an arm is eliminated under the current algorithm, it enters a ghost period—a phase during which it remains practically active (receiving the same pull probability as active arms) and is then formally eliminated at the next elimination point. Additionally, we modify the threshold in *BSC*'s Line 3 by removing the $\log(K)$ term from its additive component.

We first show that the leading term in the stochastic regret remains asymptotically unchanged, since the number of pulls during the ghost period is asymptotically smaller than the number of pulls during the active period (Lemma F.1). We then prove a variant of Lemma D.13 without the $\log(K)$ factor (Lemma F.3), which is the original source of this term in the regret bound. With these changes, the updated version of *BSC*'s Line 3 (with an appropriate choice of the constant $C$) still doesn't triggers a switch in the stochastic settings.

The $\log(K)$ factor is also removed from the adversarial regret, without affecting the asymptotic behavior. The main contribution to adversarial regret comes from the check in *BSC*'s Line 3, where we removed the $\log(K)$ term. We also need to verify that the relation between the losses and $\text{ucb}^*$ given in Equation (12) still holds. This is indeed the case, since the check in *BSC*'s Line 1 continues to be valid during the ghost period (as $i \in \mathcal{A}$ still holds).

**Lemma F.1** *Assume $G_{sto}$, we have:*

$$n(S_i^g) = O\left(n_i(\tilde{S}_i)\right)$$

**Proof:** Assume $i$ is eliminated in the $h$th elimination point and denote $S_h$ to be $S$ at that time. Let $i_h$ be the arm whose *width* crossed $2^{-h}$ in the $h$th elimination point. From the definition of *width*, we still have that its *width* is greater then $\frac{1}{2}2^{-h}$. Thus:

$$\text{width}_i(S_h) \geq \text{width}_{i_h}(S_h) \geq \frac{1}{2}2^{-h}$$

$$\text{width}_i(S_{h-1}) \leq 10\text{width}_{i_{h-1}}(S_{h-1}) \leq 10 \cdot 2^{-h+1} = 20 \cdot 2^{-h} \qquad \text{(Lemma D.10)}$$

$$40\sqrt{\frac{2\log(T)}{n_i(S_h)}} \geq \sqrt{\frac{2\log(T)}{n_i(S_{h-1})}}$$

$$1600n_i(\tilde{S}_i) \geq 1600n_i(S_{h-1}) \geq n_i(S_h) \geq n_i(S_i^g)$$

$\blacksquare$

**Lemma F.2** *Assume $G_{sto}$, for every $i \in [d]$ that is eliminated at the $h$th elimination point we have:*

$$8 \cdot 2^{-h} \leq \Delta_i \leq 320 \cdot 2^{-h}$$

**Proof:** Let $S_h$ be $S$ at the time of $h$th elimination point. From Lemma D.10:

$$\text{width}_i(S_h) \geq 2^{-h}$$

$$\text{width}_i(S_{h-1}) \leq 10 \cdot 2^{-h+1} = 20 \cdot 2^{-h}$$

Since $|S_{h-1}| \leq \left|\tilde{S}_i\right| \leq |S_h|$:

$$2^{-h} \leq \text{width}_i\left(\tilde{S}_i\right) \leq 20 \cdot 2^{-h}$$

Again from Lemma D.10:

$$8 \cdot 2^{-h} \le \Delta_i \le 320 \cdot 2^{-h}$$

∎

**Lemma F.3** *Assume $G_{sto}$, we have:*

$$\sum_i p_i^{max} \Delta_i = O(\Delta_{avg})$$

**Proof:** In the same way as Lemma D.13, we denote $\mathcal{A}$' to be the state of $\mathcal{A}$ such that $|A| = \frac{K}{2}$. In the same way we have that $\forall i \in \mathcal{A}' \quad \Delta_i \le 40\Delta_{avg}$ and:

$$\sum_i p_i^{max} \Delta_i \le \frac{\Delta_{avg}}{2} + \sum_{i \in \mathcal{A}'} p_i^{max} \Delta_i$$

Fix elimination point $h$, denote $I_h$ to be the set of arms eliminated at that point. By definition, we have for every $i \in I_h$ that $p_i^{max} \le \frac{1}{|I_h|}$. From Lemma F.2:

$$\sum_{i \in I_h} p_i^{max} \Delta_i \le \sum_{i \in I_h} \frac{1}{|I_h|} 320 \cdot 2^{-h} = 320 \cdot 2^{-h}$$

Let $h_1$ be the first elimination point in which arms from $\mathcal{A}$' are eliminated. Again from Lemma F.2, we have for some $i \in \mathcal{A}'$:

$$8 \cdot 2^{-h_1} \le \Delta_i \le 40\Delta_{avg}$$
$$2^{-h_1} \le 5\Delta_{avg}$$

This concludes to:

$$\sum_{i \in \mathcal{A}'} p_i^{max} \Delta_i = \sum_{h=h_1}^{\infty} \sum_{i \in I_h} p_i^{max} \Delta_i \le \sum_{h=h_1}^{\infty} 320 \cdot 2^{-h} = 640 \cdot 2^{-h_1} \le 3200\Delta_{avg}$$

∎

