# OpenReview forum: "Improved Best-of-Both-Worlds Regret for Bandits with Delayed Feedback"
_NeurIPS.cc/2025/Conference — NeurIPS 2025 poster_

### Official Review · Reviewer_ouSR · 2025-06-23

**Clarity:** 1
**Significance:** 3
**Originality:** 3
**Rating:** 4
**Confidence:** 3

**Summary:**

This paper proposes a best-of-both-worlds (BOBW) policy for multi-armed bandit problems with adversarial delays. The proposed policy, which switches from a sequential elimination algorithm to an external adversarial algorithm, achieves a BOBW guarantee that significantly improves upon the results obtained by the simpler algorithm in [21].

**Questions:**

1. In the current choice of ucb (lcb), it is designed to be monotonically decreasing (increasing). Could the authors point out where this monotonicity property is specifically utilized in the analysis or the algorithm's design?
2. In BSC subroutine, the confidence radius includes a $\sqrt{KT\log T}$ term. Could the authors elaborate on what would happen if this term were replaced by $\sqrt{KT}$ (that possibly remove $\log T$ term due to (3) and (4))? Which part is the essential point that introduces $\log T$ dependency?
3. RHS of Lemma C.3 includes the dependency on $t$. Should this expression include a summation over $t$?
4. The leading terms in the regret bounds appear to have quite large constant factors. Do the authors believe that these constants can be further tightened through a more refined analysis in some parts of the proof? Any thoughts on where such improvements might be possible would be appreciated.

**Ethical Concerns:**

["NO or VERY MINOR ethics concerns only"]

**Final Justification:**

I understand that the main contribution of this paper lies in establishing a BOBW guarantee under adversarial delays, with a primary focus on asymptotic regret where the order is the main target. However, the given regret bound involves sufficiently large constants, which raises concerns about its practical performance in the finite-time regime, an issue I pointed out. While the authors seem unwilling to provide empirical validation, I acknowledge that the paper is positioned as a theoretical contribution. Taking this into account, I will maintain my initial score.

**Limitations:**

yes

**Quality:**

3

**Strengths And Weaknesses:**

### Strength

The proposed policy achieves a significantly improved regret guarantee compared to previous approaches, particularly in the stochastic loss setting. Unlike the simple sequential elimination algorithm, which can perform poorly in adversarial environments, the proposed method maintains a non-zero selection probability for all arms, including those eliminated by the sequential elimination phase. By carefully designing this switching mechanism and incorporating additional procedures, the authors successfully obtain an improved BOBW regret bound. This theoretical advancement is the main contribution of the paper.

### Weakness

* Clarity issue due to incorrect links and notation.
  * $\bar{\cdot}$ is used both to denote empirical estimates (e.g., $\bar{\mu}$) and complement of sets in Line 8 of Algorithm 2, which is confusing.
  * Many internal references to algorithm steps and procedures appear to be incorrect. For example, Line 147 and Line 767. Please use precise hyperlinks or explicitly state the target (e.g., "line 3 of Procedure 3").
  * In Procedure 4, the initialization of $C_i^p$ and $E_i$ is missing (I guess the empty set and 0?). Additionally, $E_i=3\log (T)$ cannot be satisfied since $E_i$ would be natural number so that appropriate rounding using ceiling or floor functions is needed.
  * In Theorem 4.1 and throughout the manuscript (e.g., Line 236), the notation should consistently use $\log$ instead of $log$.

* Lack of numerical evaluations.

Although I understand that theoretical regret guarantees are typically the primary focus in BOBW research, and personally I do not consider the absence of experiments as a critical flaw, it would still be valuable to include empirical evaluations. This is especially important because several components of the proposed policy appear to be carefully designed to achieve tight theoretical bounds, which may introduce practical inefficiencies. A comparison with the simpler policy from [21], which has looser theoretical guarantees but may perform well in practice, would provide useful empirical insight.

* Large coefficients in regret bounds

The regret guarantees, while theoretically sound, involve very large constant factors in the leading terms, which may limit their practical relevance. A brief discussion on whether these constants can be reduced or mitigated in practice would strengthen the paper.

---

> ### Author Rebuttal · Authors · 2025-07-28
>
> **We thank the reviewer for their thoughtful and detailed feedback. Below we address the reviewer’s questions and concerns.**
>
> **Notation:**
>
> We acknowledge the clarity issues and will fix all incorrect references, inconsistent notation, and missing initializations in the final version.
>
> **Lack of numerical evaluation:**
>
> This paper is intentionally theoretical in nature, with a focus on advancing the fundamental understanding of best-of-both-worlds learning under adversarial delays. While implementation complexity and practical deployment are important directions, our goal here is to resolve core open questions regarding regret bounds and optimality. We believe that solidifying the theoretical limits is an important step toward guiding future algorithmic and empirical work.
>
> **Large constants in regret bounds:**
>
> We agree that the constants in our regret bounds are not optimal. Our main focus was on the asymptotic regret bounds, where we are near optimal.
>
> Giving improved leading constants may be a challenging future work.
>
> **The use of UCB/LCB monotonicity:**
>
> We do use the monotonicity of UCB and LCB in the analysis, specifically in line 917 and in the last inequality of line 921. These steps rely on the fact that UCBs decrease over time.
>
> **The use of log(T) in BSC's second check:**
>
> The $\log T$ term is essential to ensure that the lower bound estimate used in the BSC check stays below the actual regret in the stochastic regime. In particular, under stochastic losses, the regret can scale as $\sqrt{KT \log T}$, both in our case and more generally in Successive Elimination. For instance, setting $\Delta = \sqrt{K \log T / T}$ in the standard SE bound yields this scaling.
>
> **Lemma C.3:**
>
> We thank the reviewer for noticing this issue. The right-hand side of the lemma should indeed include a summation over $t$, and we will correct this in the final version. We stress, however, that in all places where the lemma is applied, the summation was already present and used correctly.

---

> > ### Comment · Reviewer_ouSR · 2025-08-01
> >
> > Thank you for your responses and the effort in addressing the concerns I raised.
> >
> > I understand that the main contribution of this paper lies in establishing a BOBW guarantee under adversarial delays, with a primary focus on asymptotic regret where the order is the main target. However, the given regret bound involves sufficiently large constants, which raises concerns about its practical performance in the finite-time regime, an issue I pointed out. While the authors seem unwilling to provide empirical validation, I acknowledge that the paper is positioned as a theoretical contribution. Taking this into account, I will maintain my current score.

---

### Official Review · Reviewer_pLNX · 2025-06-27

**Clarity:** 3
**Significance:** 4
**Originality:** 3
**Rating:** 5
**Confidence:** 3

**Summary:**

The paper studies bandits with delayed feedback. The feedback delays are oblivious and (to my understanding) not known to the player until the feedback arrives. The goal is a "best of both words" guarantee with log(T) regret in the stochastic case and sqrt(T) in the worst case. The paper achieves essentially optimal bounds.

**Questions:**

Maybe it makes sense to mention the interpolating setting of bandits with adversarial corruptions, which could be a natural next step for this line of work.

Also can you please clarify if the delay lengths are observed in any way before the delayed feedback arrives?

**Ethical Concerns:**

["NO or VERY MINOR ethics concerns only"]

**Final Justification:**

My positive opinion of the paper did not change.

**Limitations:**

Yes

**Paper Formatting Concerns:**

--

**Quality:**

4

**Strengths And Weaknesses:**

The paper gets optimal bounds using a non-trivial algorithm. The "adversarial" setting with delays was solved optimally using FTRL with a carefully tailored regularizer. The algorithm used here is identical once it has been decided that the environment is non-stochastic. Thus the main challenge is to get low regret in the stochastic case until this decision has been made. The algorithm proceeds in phases where the sampling rate of an arm is halved/doubled, to gradually decrease the visibility of seemingly suboptimal arms while not discarding them entirely.

I didn't read the appendix proofs in any detail but the claimed result is a significant improvement over previous (already high-quality) work and essentially optimal, and the analysis is fairly intricate, so it seems like a clear accept to me.

---

> ### Author Rebuttal · Authors · 2025-07-28
>
> **We thank the reviewer for their thoughtful evaluation and positive feedback. We address the reviewer’s comments and questions below.**
>
> **Observability of delays:**
>
> As the reviewer correctly inferred, the delays are not revealed to the learner until the corresponding feedback arrives. That is, the learner observes the loss of an action only when its loss  is revealed, without prior knowledge of the delay length. We will clarify this point more explicitly in the paper to avoid confusion.
>
> **Adversarial corruptions as future work:**
>
> We agree that the setting of adversarial corruptions presents a natural and compelling extension of this line of work. It would be interesting to explore whether the techniques developed here—especially the monitoring and adaptive switching mechanisms—can be adapted to handle adversarial corruption while preserving BoBW guarantees.

---

> > ### Comment · Reviewer_pLNX · 2025-08-07
> >
> > Thanks to the authors for the clarifications. I think we are in agreement and I will keep my positive score.

---

### Official Review · Reviewer_AA8g · 2025-07-02

**Clarity:** 3
**Significance:** 3
**Originality:** 2
**Rating:** 5
**Confidence:** 2

**Summary:**

The paper studies BoBW (stochastic + adversarial) algorithms for bandits with delayed feedback. The authors propose an algorithm that achieves $O(\sqrt{KT}+\sqrt{D})$ regret in the adversarial case (up to log-factors) and $\sum_{i\neq i^\star}log(T)/\Delta_i+\Delta_i/K\sigma_{max}$ for the stochastic case. These rates match the lower bounds for the two settings. The algorithm is active in the sense that it assumes that the algorithm is stochastic while checking if the hypothesis is true; if false, then the algorithm switches to an adversarial algorithm (similarly to the SAPO and SAO algorithms for (non-delayed) MABs).
As a stochastic algorithm, the authors employ a successive-elimination-like algorithm, but instead of eliminating arms, the probability is decreased (in some very specific way) to allow checking whether those arms become adversarial.

**Questions:**

1. Footnote 1: There is a capital letter missing
2. The main contribution of the paper is to drop the term $\Phi^*$ from the upper bounds compared to Masoudian et al. [2024]. Where does this term come from in their analysis?
3. What is the intuition behind the update of the probabilities (Procedure 4)? Lines 173-179 seem to discuss this, but I think the discussion could be improved.

**Ethical Concerns:**

["NO or VERY MINOR ethics concerns only"]

**Final Justification:**

After the rebuttals, I still find this a good theory paper that should be accepted at NeurIPS.

**Limitations:**

yes

**Quality:**

3

**Strengths And Weaknesses:**

I think this is a good paper that has non-trivial results and interesting techniques, improving on the result of Masoudian et al. [2024]. While the improvement is very technical, I think the techniques used here could be employed for similar problems.
The resulting algorithm/analysis is very technical, while prior works have simpler algorithms and analyses. It would be interesting to see if the optimal guarantees presented here are achievable by a simpler algorithm.

---

> ### Author Rebuttal · Authors · 2025-07-28
>
> **We thank the reviewer for their positive and constructive feedback. We address the reviewer’s comments and questions below.**
>
> **“It would be interesting to see if the optimal guarantees presented here are achievable by a simpler algorithm.”**
>
> Achieving optimal regret in both stochastic and adversarial regimes under delayed feedback introduces significant algorithmic and regret analysis challenges. That said, as noted in the paper, we believe that obtaining similar guarantees with a simpler algorithm is an important direction for future work.
>
> **On Footnote 1 (capitalization):**
>
> Thank you for pointing this out. We will fix the capitalization issue in the final version.
>
>
> **The term  $\Phi^*$ in Masoudian et al. [2024]:**
>
> In their work, Masoudian et al. use an explicit skipping heuristic—i.e., their algorithm actively decides to skip pulling arms in certain rounds, and the term $\Phi^*$ appears as an upper bound on the number of such skipped pulls. In contrast, once our algorithm detects that the losses are adversarial, it switches to an adversarial algorithm that handles delayed feedback, thereby avoiding the need for explicit skipping. If needed, we would be glad to expand more during the author-reviewer discussion period.
>
> **On the intuition behind the update rule (Procedure 4):**
>
> In brief, the idea is as follows: if an arm does not “behave stochastic”—that is, its observed losses appear slightly non-stochastic—we increase its probability to monitor it more closely. If the arm behaves stochastically throughout a phase, we can safely reduce its probability. This adaptive adjustment enables the algorithm to maintain control over regret in both stochastic and adversarial regimes.
>
> We agree that the current discussion can be improved and convey better the intuition behind the update mechanism. While Lines 158–170 describe the update rule, the actual motivation becomes more apparent only in the analysis of the adversarial case (Lines 282–286). We will revise the earlier section to make this intuition explicit.

---

> > ### Comment · Reviewer_AA8g · 2025-08-01
> >
> > Thanks for the reply.
> > I find the answers to be satisfactory, and I still think this to be a good paper worth accepting at NeurIPS

---

### Official Review · Reviewer_9pBg · 2025-07-02

**Clarity:** 2
**Significance:** 2
**Originality:** 2
**Rating:** 4
**Confidence:** 3

**Summary:**

This paper presents a new algorithm for the multi-armed bandit problem with adversarially chosen delays. The authors achieve optimal performance in both stochastic and adversarial settings and advance the Best-of-Both-Worlds (BoBW) framework. The proposed algorithm adapts between regimes by starting with a stochastic strategy and switching to an adversarial one when non-stochastic behavior is detected. The regret bounds matches the known lower bounds up to log factors. They improve on prior BoBW approaches and and outperform existing stochastic-specific algorithms by reducing delay-related regret terms.

**Questions:**

1) I do not get Line 253 and Appendix F. Either integrate the "slight modification" from Appendix F into the main algorithm and results to present the stronger bounds without the $\log(K)$ factor, or provide a clearer justification in the main body why it is given only in the appendix.

2) What is ALG? The core algorithm, Delayed SAPO (Algorithm 2 and 5), integrates an "external algorithm for adversarial settings, ALG". While Zimmert and Seldin is mentioned as an example (Line 289, Corollary 5.4), the paper does not explicitly detail the properties or assumptions required for ALG (e.g., its specific regret bounds, whether it must handle unknown delays, its computational overhead, or specific feedback requirements) for the overall theoretical guarantees (Theorem 5.1) to hold. Explicitly state the necessary properties or assumptions required of the external ALG algorithm (e.g., its regret performance in delayed feedback) in the main paper to make the algorithm's dependencies clear.

3) I think you can elevate the "Remark" about the "general trick" for $\Delta_{avg}$ dependence (instead of $\Delta_{max}$) in Lines 246-250 from the proof sketch to a more prominent section to emphasize it since it seems as a broader methodological contribution.

**Ethical Concerns:**

["NO or VERY MINOR ethics concerns only"]

**Final Justification:**

I am keeping my score, which remains positive.

Authors clarified the placement of the stronger bound in Appendix for simplicity and will revise for clarity. Authors confirmed ALG is an external adversarial algorithm, and its properties do not impose specific restrictions beyond its own guarantees, allowing any equivalent ALG. Authors will highlight the "General trick" technique more prominently.

Lack of empirical evaluation, implementation complexity, and practical deployment remains an absence, but it's an accepted limitation for a primarily theoretical paper focused on fundamental regret bounds and optimality. This practical exploration is deemed future work. Exploration of non-adversarial delay models is acknowledged as interesting for future research, but outside the scope of this theoretical work.

The core theoretical contribution of matching lower bounds is extremely strong and heavily weighted. While practical considerations are important, their absence is appropriate for the paper's theoretical scope.

**Limitations:**

yes

**Paper Formatting Concerns:**

no major issues

**Quality:**

2

**Strengths And Weaknesses:**

# Strengths

1) The regret bounds are tight in both the stochastic and adversarial settings with adversarial delays, matching known lower bounds up to logarithmic factors.

2) It improves upon prior best-of-both-worlds (BoBW) and stochastic-specific algorithms by tightening the delay-dependent term in the stochastic regime, achieving a regret bound with improved dependence.

3) The algorithm dynamically switches between stochastic and adversarial strategies based on observed feedback, and performs across a wide range of environments without prior knowledge of the regime.

4) The design introduces refined elimination criteria and a novel subroutine for managing eliminated arms (EAP).

5) The theoretical analysis is comprehensive.

# Weaknesses

1) No empirical evaluation. The absence of experiments makes it difficult to assess and validate the algorithm’s practical performance, robustness, and computational efficiency.

2) The introduced techniques increase implementation complexity and may pose challenges in practice despite their theoretical justification.

3) The paper focuses on theory with minimal consideeration to practical deployment (such as runtime cost, ease of use, or hyperparameter tuning).

4) The work assumes fully adversarial delays and does not explore common settings like i.i.d. or bounded delays, which might lead to stronger results.

---

> ### Author Rebuttal · Authors · 2025-07-28
>
> **We thank the reviewer for their thoughtful and constructive feedback. Below we address the reviewer’s specific concerns and questions.**
>
> **On the lack of empirical evaluation and practical considerations:**
>
> This paper is intentionally theoretical in nature, with a focus on advancing the fundamental understanding of the optimal regret for best-of-both-worlds learning under adversarial delays. While implementation complexity and practical deployment are important directions, our goal here is to resolve core open questions regarding regret bounds and optimality. We believe that understanding the theoretical regret bounds is an important step toward guiding future algorithmic and empirical work.
>
> **On non-adversarial delay models (e.g.,stochastic delays):**
>
> We agree that other structured delay models, such as i.i.d. or bounded delays, are of interest. In fact, we explicitly mention this (i.i.d. or payoff-dependent delays) as a direction for future work in the paper.
>
> **On the “slight modification” in Appendix F:**
>
> We placed the stronger bound in Appendix F to reduce the complexity of the already intricate main algorithm. This rationale is explained at the first point of reference (Line 186), but we acknowledge that it is not reiterated at the second mention (Line 253), which may have caused confusion. We will revise the manuscript to ensure that the motivation is clearly communicated at both points of reference.
>
> **On the role of ALG in the algorithm:**
>
> The core algorithm is defined in terms of an external adversarial algorithm, ALG, and we do not impose any restrictions on ALG in the algorithm itself. In Theorem 5.1, we assume only that ALG achieves some regret bound R_{ALG}​ in terms of T, K, D and possibly d_max and sigma_max, when handling unknown delays.  We later instantiate ALG with a specific algorithm in Corollary 5.4 to derive an explicit regret bound. We view our ability to incorporate any ALG as a benefit of our approach.
>
> We will revise the presentation to make these assumptions and their roles in the analysis more explicit and easier to follow.
>
> **On the “general trick” for handling delay dependence (Lines 246–250):**
>
> We appreciate the suggestion to highlight this technique more prominently. It already appears as a standalone “Remark” block in the proof sketch to emphasize its generality, but we agree it is a broadly useful idea and will revise the presentation to make this clearer and more visible.

---

> > ### Comment · Reviewer_9pBg · 2025-08-05
> >
> > I appreciate the authors' rebuttal and have some follow up comments.
> >
> > - While I acknowledge that the numerical experiments are not necessary for a theoretically focused work, I still believe the investigation on the computational complexity to be a bit lacking. Although achieving optimal regret bounds are important, their implementation feasibility is an important point of discussion.
> >
> > - You clarified that ALG is defined as an "external adversarial algorithm" and referenced Zimmert and Seldin in Corollary 5.4 for its regret bound. To enhance understanding, could you specify what properties an external algorithm (ALG) must possess to allow the overall "Delayed SAPO Algorithm" to achieve the "optimal $\tilde{O}(\sqrt{KT} + \sqrt{D})$ regret" in the adversarial regime, as claimed in your abstract?

---

> > > ### Author Response · Authors · 2025-08-06
> > >
> > > * We agree with the reviewer that an empirical evaluation could be insightful and we leave it for future research, and it is outside the scope of our theoretical work.
> > >
> > > * Regarding the second point, as stated in Theorem 5.1, our regret bound is $\tilde{O}(\sqrt{TK} + \sigma_{max} + \mathcal{R}{\tiny ALG})$ where $\mathcal{R}{\tiny ALG}$ is the regret guarantee of the external algorithm $ALG$. This theorem applies to any online learning algorithm for delayed adversarial losses. In the case of using Zimmert and Seldin [39] as $ALG$, the regret guarantee is $\mathcal{R}{\tiny ALG} \leq \sqrt{TK} + \sqrt{D}$. Combining this with the observation that $\sigma_{max} \leq \sqrt{D}$ gives the desired regret bound $\tilde{O}(\sqrt{KT} + \sqrt{D})$. We remark that there are additional algorithms with equivalent regret bound (such as [13] in our reference list) and they all can be used to derive our result.

---

### Decision · Program_Chairs · 2025-09-17

**Decision:**

Accept (poster)

**Comment:**

This paper proposes a best-of-both-worlds (BoBW) algorithm for multi-armed bandits with adversarial delays. As proved by the authors, this algorithm can achieve near-optimal regret bounds in both the stochastic and adversarial settings (about the distribution of loss functions, instead of the delay), which is better than the existing best BoBW algorithm in Masoudian et al. [21].

All reviewers agree that this paper provides a strong theoretical contribution for multi-armed bandits with adversarial delays, and give positive scores before and after the rebuttal period. There also exist some weaknesses, including the lack of empirical evaluation, questions about implementation complexity, and large constants in the regret bounds. Nonetheless, the theoretical advances are substantial and impactful. Thus, I recommend accepting this paper.